



**A comparison of gap-filling algorithms for eddy covariance**
**fluxes and their drivers**
Atbin Mahabbati[1], Jason Beringer[1], Matthias Leopold[1], Ian McHugh[2], James Cleverly[3], Peter Isaac[4],
Azizallah Izady[5]
[1] School of Agriculture and Environment, The University of Western Australia, 35 Stirling Hwy,
Crawley, Perth WA, 6009, Australia
[2] School of Ecosystem and Forest Sciences, The University of Melbourne, Richmond, VIC, 3121,
Australia
[3] School of Life Sciences University of Technology Sydney Broadway NSW  2007
[4] OzFlux Central Node, TERN Ecosystem Processes, Melbourne, VIC 3159, Australia
[5] Water Research Center, Sultan Qaboos University, Muscat, Oman
*Correspondence to:* Atbin Mahabbati (atbin.m@hotmail.com)

## Abstract

17        The errors and uncertainties associated with gap-filling algorithms of water, carbon and
energy fluxes data, have always been one of the prominent challenges of the global network of
microclimatological tower sites that use eddy covariance (EC) technique. To address this concern, and
find more efficient gap-filling algorithms, we reviewed eight algorithms to estimate missing values of
environmental drivers, and separately three major fluxes in EC time series. We then examined the
performance of mentioned algorithms for different gap-filling scenarios utilising data from five
OzFlux Network towers during 2013.  The objectives of this research were a) to evaluate the impact
of training and testing window lengths on the performance of each algorithm; b) to compare the
performance of traditional and new gap-filling techniques for the EC data, for fluxes and their
corresponding meteorological drivers. The performance of algorithms was evaluated by generating
nine different training-testing window lengths, ranging from a day to 365 days. In each scenario, the
gaps covered the data for the entirety of 2013 by consecutively repeating them, where, in each step,
values were modelled by using earlier window data. After running each scenario, a variety of
statistical metrics was used to evaluate the performance of the algorithms. The algorithms showed
different levels of sensitivity to training-testing windows; The Prophet Forecast Model (FBP) revealed
the most sensitivity, whilst the performance of artificial neural networks (ANNs), for instance, did not
vary considerably by changing the window length. The performance of the algorithms generally
decreased with increasing training-testing window length, yet the differences were not considerable
for the windows smaller than 60 days. Gap-filling of the environmental drivers showed there was not
a significant difference amongst the algorithms, the linear algorithms showed slight superiority over



those of machine learning (ML), except the random forest algorithm estimating the ground heat flux (RMSEs of 30.17 and 34.93 for RF and CLR respectively). For the major fluxes, though, ML algorithms showed superiority (9 % less RMSE on average), except the Support Vector Regression (SVR), which provided significant bias in its estimations. Even though ANNs, random forest (RF) and extreme gradient boost (XGB) showed close performance in gap-filling of the major fluxes, RF provided more consistent results with less bias, relatively. The results indicated that there is no single algorithm which outperforms in all situations and therefore, but RF is a potential alternative for the ANNs as regards flux gap-filling.

## 1. Introduction

To address the global challenges of climatological and ecological changes, environmental scientists and policymakers are demanding data that are continuous in time and space. Besides, there is a need for quantifying and reducing uncertainties in such data, including observations of carbon, water and energy exchanges that are crucial components in national/international flux networks and global earth observing systems. Satellites partially fill this gap as they provide excellent spatial coverage but at a limited temporal resolution, and not measured at the point. As such, high-quality long-term site observations of ecosystem process and fluxes are needed that are continuous in time and space. The global eddy covariance (EC) flux tower networks (FLUXNET), consisted of its regional counterparts (i.e. AmeriFlux, EUROFLUX, OzFlux, etc.), was established in the late 1990s to address the global demand for such information (Beringer et al., 2016a; Hollinger et al., 1999; Tenhunen et al., 1998). Despite the capability of EC to frequently validate process modelling analyses, field surveys and remote sensing assessments (Hagen et al., 2006), there are some serious concerns regarding the challenges associated with the technique, e.g. data gaps and uncertainties. Hence, filling data gaps and reducing uncertainties through better gap-filling techniques are highly needed.

Even though the EC is a common technique to measure fluxes of carbon, water and energy, there are some challenges in providing robust, high-quality continuous observations. One of the challenges regarding the technique, and therefore, the flux networks, is addressing data gaps and the uncertainties associated with the gap-filling process, mainly when the gap windows are long (longer than 12 consecutive days, as described by (Moffat et al., 2007)). These gaps happen very often due to a variety of reasons, such as values out of range, spike detection or manual exclusion of date and time ranges, instrument or power failure, herbivores, fire, eagles nests, cows, lightning, researchers on leave, etc. (Beringer et al., 2016b). Since EC flux towers are often located in harsh climates, their data are more susceptible to adverse weather (i.e. rain conditions), and they sometimes prevent quick access to sites for repair and maintenance. As a result, this issue can, in turn, produce gaps which might be relatively long (Isaac et al., 2017), and thus, problematic as follows. Firstly, loss of data is considered a threat to scientific studies depending on the missing data quantity, pattern, mechanism and nature (Altman and Bland, 2007; Molenberghs et al., 2014; Tannenbaum, 2010). That is because using an incomplete dataset might lead to biased, invalid and unreliable results (Allison, 2000; Kang,





2013; Little, 2002). Second, continuous gap-filled data are required to calculate the annual or monthly
budgets of carbon or water balance components (Hutley et al., 2005).

Other than the challenges caused by missing data, there are several sources of errors and
uncertainties in the EC technique. Firstly, random error is associated with the stochastic nature of
turbulence, associated sampling errors (incomplete sampling of large eddies, uncertainty in the
calculated covariance between the vertical wind velocity and the scalar of interest), instrument errors,
and footprint variability (Aubinet et al., 2012). For instance, Dragoni et al. (2007) analysed an EC-based
data of Morgan-Monroe State Forest for eight years (1999-2006) and assessed that instrument
uncertainty was equal to 3 % of the total annual NEE. Another primary source of uncertainty in EC
measurements is systematic errors that are usually caused by methodological challenges and
instrument calibration problems (e.g. sonic anemometer errors, spikes, gas analyser errors, etc.).
Finally, one of the sources of uncertainties is data processing, especially data gap-filling (Isaac et al.,
2017; Moffat et al., 2007; Richardson et al., 2012; Richardson and Hollinger, 2007).

There are several uncertainties pertaining to gap-filling of missing values, including
measurement uncertainty (Richardson and Hollinger, 2007), lengths and timing the gaps (Falge et al.,
2001; Richardson and Hollinger, 2007) and the particular gap-filling algorithm that is used (Falge et
al., 2001; Moffat et al., 2007).  However, there are two dominant issues of long data gaps and the choice
of a particular gap-filling algorithm (Aubinet et al., 2012). Firstly, long gaps can significantly increase
the total amount of uncertainty as the ecosystem behaviour might change because of different
agricultural periods or phenological phases (e.g. growing season, harvest period, bushfire, etc.). And
thereby show different responses under similar meteorological conditions (Aubinet et al., 2012; Isaac
et al., 2017; Richardson and Hollinger, 2007). Consequently, the period in which a long gap happens
is essential. For example, research undertook by Richardson & Hollinger (2007) on data from a range
of FLUXNET sites revealed that a week data gap during spring green-up in a forest led to a higher
uncertainty over a three-week gap period during winter. Second, each gap-filling algorithm has its
strengths and weaknesses; for instance, Moffat et al. (2007) compared a couple of different commonly-
used gap-filling algorithms. They found that there was not a significant difference between the
performances of the algorithms with "good" reliability based on analysis of variance of RMSE.
Besides, the overall gap-filling uncertainty was within ±25 g C m$^{-2}$ yr$^{-1}$ for most of the proper
algorithms, whereas, the other algorithms generated higher uncertainties of up to ±75 g C m$^{-2}$ yr$^{-1}$. This
result is similar to the findings of Richardson & Hollinger (2007)  who found uncertainties of up to
±30 g C m$^{-2}$ yr$^{-1}$ for long gaps by appropriate algorithms. Considering that the data provided by EC
tower networks are of use for research, government and policymakers, robust gap-filling is a need to
quantify and reduce uncertainties in flux estimations.

To manage the missing data problem, several methods have been typically used to fill data
gaps in both fluxes and their meteorological drivers. Due to computational constraints of complex





algorithms, early works to impute EC data gaps used interpolation methods based mostly on linear
regression or temporal autocorrelation (Falge et al., 2001; Lee et al., 1999). These approaches were
replaced quickly by more sophisticated methods such as non-linear regressions (Barr et al., 2004; Falge
et al., 2001; Moffat et al., 2007; Richardson et al., 2006); lookup tables (Falge et al., 2001; Law et al.,
2002; Zhao and Huang, 2015); artificial neural networks (ANNs) (Aubinet et al., 1999; Beringer et al.,
2016a; Cleverly et al., 2013; Hagen et al., 2006; Isaac et al., 2017; Kunwor et al., 2017; Moffat et al., 2007;
Papale and Valentini, 2003; Pilegaard et al., 2001; Staebler, 1999); mean diurnal variation (Falge et al.,
2001; Moffat et al., 2007; Zhao and Huang, 2015), multiple imputations (Hui et al., 2004; Moffat et al.,
2007), etc. Each of these methods has its pros and cons as follows: a) Interpolation methods such as
the Mean Diurnal Variation (MDV), do not need any drivers, yet, their accuracy is lower than other
approaches (Aubinet et al., 2012). Moreover, this method may provide biased results on extremely
clear or cloudy days (Falge et al., 2001). MDV is not recommended when a gap is longer than two
weeks, for it cannot consider the non-linear relations between the drivers and the flux, and thus leads
to a high level of uncertainty (Falge et al., 2001). And b) The Lookup table, especially its modified
version, Marginal Distribution Sampling (MDS), has provided performance close to ANNs, and are
more reliable and consistent than the other algorithms so far. Hence, MDS was chosen as one of the
standard gap-filling methods in EUROFLUX (Aubinet et al., 2012). Nevertheless, one of the concerns
regarding this algorithm is that the independent variables, here meteorological drivers, might be auto-
correlated. c) ANNs have commonly been used to gap-fill EC fluxes since 2000 and because of their
robust and consistent results are considered as a standard gap-filling algorithm in several networks,
e.g. ICOS, FLUXNET, OzFlux, etc. (Aubinet et al., 2012; Beringer et al., 2017; Isaac et al., 2017). Despite
their reliable performance, ANNs –and generally all other ML algorithms- face some challenges. Over-
fitting, for instance, is a big concern and can happen when the number of degrees of freedom is high,
while the training window is not long enough respectively, or the quality of the training dataset is
low. This challenge becomes acute when the gaps happen within a period when the ecosystem
behaviour is changing and thereby showing different response under similar meteorological
conditions. Furthermore, there is a desire to have the training windows short so that the algorithm
can track the ecosystem behaviour shift. Yet, this increases the risk of over-fitting depending on the
algorithm. In other words, the training window length should be neither too short to cause over-
fitting, and nor too long to lead algorithms to ignore ecological condition changes. Besides, long gaps
are considered as one of the primary uncertainty sources of $CO_2$ flux in the FLUXNET (Aubinet et al.,
2012). As a result, studying the effects of the gap lengths, as well as the window length whereby an
algorithm is trained are both critical challenges associated with the environmental data gap-filling.

Apart from the limitations and disadvantages of the mentioned algorithms, gap-filling of fluxes
(i.e. NEE) experiences some other challenges that make it necessary to find or develop new gap-filling
algorithms. That is because the current methods are not flexible enough to perform well in special
occasions or extreme values (Kunwor et al., 2017), and there is almost no room to optimise them to
improve their outcome (Moffat et al., 2007). Moreover, even using the best available algorithm, such



as ANNs, the model (gap-filling) uncertainty still accounts for a sizable proportion of the total
uncertainties, especially when the gaps are relatively long. Since the 2000s when MDS and ANNs were
chosen as the most reliable gap-filling methods for EC flux observations, many new ML and
optimisation algorithms have been developed and used in varieties of scientific fields. Some of which
have shown superiority over ANNs, either individually or as a part of a hybrid or ensemble model,
e.g. (Gani et al., 2016). As a result, comparing the cutting-edge algorithms with the current standard
ones can show whether there is any room to improve the gap-filling process within the field.
According to the concerns mentioned above, this paper had two objectives. a) To find out the impact
of different window lengths on the performance of each algorithm. And b) evaluate the performance
of traditional and new gap-filling techniques for the OzFlux Network, separately for fluxes and their
meteorological drivers, particularly soil moisture, for this has always been a challenging variable to
gap-fill for a couple of reasons, such as of the biology and heterogeneity of soil parameters. To address
these objectives, we utilised eight different algorithms (Extreme Gradient Boost (XGB), Random Forest
Algorithm (RF), Artificial Neural Networks (ANNs), Classic Linear Regression (CLR), Support Vector
Regression (SVR), Elastic net regularisation (ELN), Panel Data (PD) and Prophet Forecast Model
(FBP)) to fill the gaps of environmental drivers and the major fluxes. We then assessed their relative
performance to evaluate potentially better ways to fill EC flux data. To test the approaches, we used
five flux towers from the OzFlux network. To evaluate the performance of these algorithms, nine
scenarios for gaps were planned – from a day to a whole year - and applied to the datasets, and
different common performance metrics (e.g. RMSE, MBE, etc.), as well as visual graphs were used.

## 2. Materials and methods


To address the first objective of this research, data of nine different window lengths were
considered to train and test the algorithms, i.e. 1, 5, 10, 20, 30, 60, 90, 180 and 365 days. To address the
second objective, we chose eight different algorithms to fill the gaps, including a wide variety of
different approaches, e.g. from a simple algorithm like CLR to cutting-edge ML algorithms, such as
XGB. The data used in this paper came from five EC towers of the OzFlux Network, i.e. Alice Springs
Mulga, Calperum, Gingin, Howard Springs and Tumbarumba form 2011 to 2013, with a time
resolution of 30 minutes. Additionally, data coming from three additional sources outside of the
network were also used as ancillary data to help the algorithms fill the gaps of environmental drivers.

### 2.1. Data
The data used for this research came from OzFlux, which is the regional Australian and New
Zealand flux tower network that aims to provide a continental-scale national research facility to
monitor and assess Australia's terrestrial biosphere and climate (Beringer et al., 2016a). As described
in (Isaac et al., 2017), all OzFlux towers continuously measure and record 28 environmental features
at resolutions up to 10 Hz, and use a 30 min averaging period, with a few exceptions (data are available
from (http://data.ozflux.org.au/portal). Besides, the network acquires additional data from the
Australian Bureau of Meteorology (BoM), the European Centre for Medium-Range Weather

Forecasting (ECMWF), and the Moderate Resolution Imaging Spectroradiometer (MODIS) on the TERRA and AQUA satellites (Isaac et al., 2017). These additional data, also known as ancillary data, provide alternative data for gap-filling flux tower datasets (Isaac et al., 2017). As explained in (Isaac et al., 2017), OzFlux uses the BoM automated weather station (AWS) datasets to gap-fill the meteorological data, the BoM weather forecasting model (ACCESS-R) for radiation and soil data from 2011 onward, and MODIS MOD13Q1 for Normalised Difference Vegetation Index (NDVI) and Enhanced Vegetation Index (EVI). Moreover, the data provided by BIOS2, a physically-based model-data integration environment for tracking Australian carbon and water (Haverd et al., 2015), were also used as another ancillary source for varieties of environmental features. Current ACCESS-R and MODIS data are available from the BoM OPeNDAP (http://www.opendap.org/) server and TERN-AusCover data (http://www.auscover.org.au/), respectively.

The datasets were used in this research came from five towers amongst the OzFlux Network between 2011 and 2013, each representative of a different climate and land cover of Australian ecological conditions; i.e. Alice Springs Mulga: Tropical and Subtropical Desert, Calperum: steppe, Gingin: Mediterranean, Howard Springs: Tropical Savanna, Tumbarumba: Oceanic (Table 1) (Beringer et al. 2016). The datasets included 15 meteorological drivers as well as three major fluxes recorded (Table 2) based upon EC technique at a 30-minute temporal resolution, except for Tumbarumba, which was hourly. Additionally, relevant ancillary datasets for the mentioned towers were used to follow the OzFlux Network gap-filling protocol. Each dataset was quality checked at three levels based on the OzFlux Network protocol described in (Isaac et al., 2017) and applied using PyFluxPro ver. 0.9.2. To address the underestimation of canopy respiration by EC measurements at night, we used the CPD method of (Barr et al., 2013) to reject nightly records when the friction velocity fell below the threshold value of each site. After dismissing the inappropriate measurements, overall coverage of 72-88 % and 21-48 % were achieved for diurnal and nocturnal records, respectively.

*Table 1. The information of the five towers that their data were used, including their name, location, dominant species and climate.*

| Site | Location | Species | Climate | Latitude, Longitude (degree) |
|---|---|---|---|---|
| Alice Springs Mulga [AU-ASM] | Pine Hill cattle station, near Alice Springs, Northern Territory | Semi-arid mulga (Acacia aneura) ecosystem | Tropical and Subtropical Desert Climate (Bwh) | -22.2828° N, 133.2493° E |
| Calperum [AU-Cpr] | Calperum Station, 25 km NW of Renmark, South Australia | Recovering Mallee woodland | Steppe Climate (Bsk) | -34.0027° N, 140.5877° E |
| Gingin [AU-Gin] | Swan Coastal Plain 70 km north of Perth, Western Australia | Coastal heath Banksia woodland | Mediterranean Climate (Csa) | -31.3764° N, 115.7139° E |
| Howard Springs [AU-How] | E of Darwin, NT | Tropical savanna (wet) | Tropical Savanna Climate (Aw) | -12.4943° N, 131.1523° E |
| Tumbarumba [AU-Tum] | Near Tumbarumba, NSW | Wet temperate sclerophyll eucalypt | Oceanic climate (Cfb) | -35.6566° N, 148.1517° E |




*Table 2. List of variables and their units used in this research, including the three main fluxes and their environmental drivers.*

| List of variables | Units |
|---|---|
| **Drivers:** | |
| Ah | Absolute Humidity (g m$^{-3}$) |
| Fa | Available energy (W m$^{-2}$) |
| Fg | Ground heat flux (W m$^{-2}$) |
| Fld | Downwelling long-wave radiation (W m$^{-2}$) |
| Flu | Upwelling long-wave radiation (W m$^{-2}$) |
| Fn | Net radiation (W m$^{-2}$) |
| Fsd | Downwelling short-wave radiation (W m$^{-2}$) |
| Fsu | Upwelling short-wave radiation (W m$^{-2}$) |
| ps | Surface pressure (kPa) |
| Sws | Soil water content (m m$^{-1}$) |
| Ta | Air temperature (C) |
| Ts | Soil temperature (C) |
| Ws | Wind speed (m s$^{-1}$) |
| Wd | Wind direction (deg) |
| Precip | Precipitation (mm) |
| **Fluxes:** | |
| Fc | CO$_2$ flux ($\mu$mol m$^{-2}$ s$^{-1}$) |
| Fh | Sensible heat flux (W m$^{-2}$) |
| Fe | Latent heat flux (W m$^{-2}$) |


The datasets whereby each environmental variable was gap-filled are shown in Table 3. For each of
these variables, the same variable of the ancillary source was used to fill the gaps. For instance, to gap-
fill Ah, the Ah records of AWS, ACCESS-R and BIOS2 were used. To gap-fill the missing values of
fluxes, i.e. Fc, Fh and Fe, eight drivers were used as follows: Ta, Ws, Sws, Fg, VPD, Fn, q and Ts based
on trial and error. Different libraries of Python Programming Language (ver. 3.6.4) were utilised for
training and testing the algorithms, i.e. xgboost for XGB, fbprophet for FBP, statsmodels for PD and
sklearn for the rest of algorithms. Each algorithm was tuned up individually using gird search, and
the number of nodes, layers, irritations, etc. were chosen therefor.


*Table 3. The ancillary sources whereby each environmental driver was gap-filled.*

| List of variables (y) | Ancillary Source |
|---|---|
| **Drivers:** | |
| Ah | AWS, ACCESS-R, BIOS2 |
| Fa | ACCESS-R, BIOS2 |
| Fg | ACCESS-R, BIOS2 |
| Fld | ACCESS-R, BIOS2 |
| Flu | ACCESS-R, BIOS2 |
| Fn | ACCESS-R, BIOS2 |
| Fsd | ACCESS-R, BIOS2 |
| Fsu | ACCESS-R, BIOS2 |
| ps | AWS, ACCESS-R |
| Sws | ACCESS-R, BIOS2 |



| | |
|---|---|
| Ta | AWS, ACCESS-R, BIOS2 |
| Ts | ACCESS-R, BIOS2 |
| Ws | AWS, ACCESS-R |
| Wd | AWS, ACCESS-R |
| Precip | AWS, ACCESS-R, BIOS2 |

### 2.2. Gap-filling algorithms

Eight imputation algorithms for estimating 15 environmental drivers and 3 major fluxes were picked out to make the comparison. These algorithms were used in a way that a variety of approaches were tested, from a standard method like ANNs, to the newer algorithms which rarely or never been used in the field, such as Extreme Gradient Boosting and panel data. According to the literature, since the performance of ANNs had been either equal or slightly better than MDS (Kim et al., 2020; Moffat et al., 2007), we did not use MDS.

**Artificial Neural Networks (ANN)**

Rooted in the 1950s, artificial neural networks are ML methods inspired by biological neural networks and are classified as supervised learning methods (Dreyfus, 1990; Farley and Clark, 1954). ANN work based on several connected units called nodes, which are used to mimic the functionality of a neuron in an animal brain by sending and receiving signals to other nodes. The ANN technique used in this paper was Multi-layer Perceptron regressor, which optimises the squared-loss using stochastic gradient descent. Sklearn.neural_network.MLPRegressor was used to apply this method in Python, and its hyperparameters were 800 and 500 for "hidden_layer_sizes" and "max_iter", respectively based on grid search. ANN are one of the current standard approaches for gap-filling in FLUXNET and in this research were picked out as a performance reference for other algorithms.

**Classical Linear Regression (CLR)**

A classical linear regression is an equation developed to estimate the value of the dependent variable (y) based on independent values ($x_i$). In contrast, each $x_i$ has its specific coefficient and an overall intercept value. In this method, these coefficients are determined by minimising the squared residuals (errors) of estimated vs observed values, called least squares. A CLR algorithm can be formulated as follows (Freedman, 2009):

$$y = \alpha + \beta_1 X_1 + \beta_2 X_2 + \beta_3 X_3 + \ldots + \beta_i X_i + \varepsilon \tag{1}$$

where y is the dependent variable, $\alpha$ is the interception, Xs are independent variables, and $\beta_i$ is coefficient of Xi, and $\varepsilon$ is the error term. We chose this algorithm as a baseline to find out how better more complicated algorithms can estimate dependent variables comparatively.



**Random Forests (RF)**

Random forest, a supervised ML algorithm, used for both classification and regression, consists of multiple trees constructed systematically by pseudorandomly selecting subsets of components of the feature vector, that is, trees constructed in randomly chosen subspaces (Ho, 1998). RF algorithm has been developed to control the overcome over-fitting problem, a commonplace limitation of its preceding decision tree-based methods (Ho, 1995, 1998). Sklearn.ensemble.RandomForestRegressor was used to apply this method in Python, and the hyperparameters used were 5 and 1000 for "max_depth" and "n_estimators", respectively based on grid search.

**Support Vector Regression (SVR)**

As a non-linear method, support vector regression was developed based on Vanpik's concept of support vectors theory (Drucker et al., 1997). An SVR algorithm is trained by trying to solve the following problem:

minimise $\frac{1}{2}\|w\|^2$

subject to $\begin{pmatrix} y_i - \langle w, x_i \rangle - b \leq \varepsilon, \\ \langle w, x_i \rangle + b - y_i \leq \varepsilon, \end{pmatrix}$

where $x_i$ and $y_i$ are training sample and target value in a row. The inner product plus intercept $\langle w, x_i \rangle + b$ is the prediction for that sample, and $\varepsilon$ is a free parameter that serves as a threshold. sklearn.svm.SVR was used to apply this method in Python, and the hyperparameters that used were 1 and 0.001 for "C" and "gamma", respectively based on grid search.

**Elastic net regularisation (ELN)**

The elastic net is a linear regularised regression method that exerts small amounts of bias by adding two penalty components to the regressed line to decline the coefficients of independent variables and thus, provides better long-term predictions. Given that these two penalty components come from ridge regression and LASSO, the elastic net is considered as a hybrid model consists of ridge and LASSO regressions, overcoming the limitations of both. The estimates from the ELN method can be formulated as below (Zou and Hastie, 2005):

$$\hat{\beta}(elastic\ net) = \frac{\left(|\hat{\beta}(OLS)| - \lambda_1/2\right)}{1 + \lambda_2} sgn\{\hat{\beta}(OLS)\} \tag{2}$$

where $\hat{\beta}$ is the coefficient of each ELN independent variable, $\lambda_1$ and $\lambda_2$ are penalty coefficients of LASSO and ridge regression respectively, $\hat{\beta}(OLS)$ is the coefficient of an independent variable calculated based on ordinary least squares, and *sgn* stands for the sign function:



$$sgn(x) = \begin{cases} 1 & x > 0 \\ 0 & x = 0 \\ -1 & x < 0 \end{cases} \tag{3}$$


The ELN regression is good at addressing situations when the training datasets have small samples
or when there are correlations between parameters. sklearn.linear_model.ElasticNet was used to
apply this method in Python, and the hyperparameters used were as follows: {'alpha': 0.01,
'fit_intercept': True, 'max_iter': 5000, 'normalize': False} based on grid search.

**Panel data (PD)**
Panel data is a multidimensional statistical method, mainly used in econometrics to analyse
datasets, which involve time series of observations amongst individual cross-sections (Baltagi, 1995)
usually based on ordinary least squares (OLS) or generalised least squares (GLS). A two-way panel
data model consists of two extra components above a CLR as follows (Baltagi, 1995; Hsiao et al., 2002;
Wooldridge, 2008):

$$y_{it} = \alpha + \beta X_{it} + u_{it} \qquad i = 1, 2, ..., N ; \;\; t = 1, 2, ..., T \tag{4}$$

$$y_{it} = \alpha + \beta X_{it} + \mu_i + \lambda_t \tag{5}$$

where i and t denote the cross-section and time series dimension in a row, y is a dependent-variable
vector, X is an independent variable matrix, $\alpha$ is a scalar, $\beta$ is the coefficient of the independent-
variable matrix, $\mu_i$ is the unobservable individual-specific effect, and $\lambda_t$ is the unobservable time-
specific effect. Panel data abilities to provide a holistic analysis of different individuals, as well as
determining the specific impact of every single time caused its superiority over CLR.
**Extreme Gradient Boost (XGB)**
Extreme gradient boost is a reinforced method of Gradient Boost introduced in 1999 that works based
on parallel boosted decision trees and similar to RF can be used for a variety of data processing
purposes including classification and regression (Friedman, 2002; Jerome H. Friedman, 2001; Ye et al.,
2009). XGB method is resistive to over-fitting and provides a robust, portable and scalable algorithm
for large-scale boosting decision-trees-based techniques.
sklearn.ensemble.GradientBoostingRegressor was used to apply this method in Python, and its
hyperparameters were chosen based on grid search as follows: {'learning_rate': 0.001, 'max_depth': 8,
'reg_alpha': 0.1, 'subsample': 0.5}.

**The Prophet Forecasting Model (FBP)**
The Prophet Forecasting Model, also known as "prophet", is a time series forecasting model
developed by Facebook to manage the common features of business time series and designed to have





intuitive parameters that can be adjusted without knowing the details of underlying model (Taylor
and Letham, 2017). A decomposable time series model was used (Harvey and Peters, 1990) to develop
this model, with three main components: trend, seasonality, and holidays as the equation below
(Taylor and Letham, 2018):

$$y(t) = g(t) + s(t) + h(t) \qquad (6)$$


where $g(t)$ is the trend function, which models mom-periodic changes, $s(t)$ is a function to represent
periodic changes, e.g. seasonality, and $h(t)$ assesses the effects of potential anomalies which occur over
one or more days, e.g. holidays.

### 2.3. *The gap scenarios*

To find out the effect of gap size on the performance of our gap-filling algorithms, we trained
each of them using nine different window lengths (i.e. 1, 5, 10, 20, 30, 60, 90, 180 and 365 days). The
gap size for each trained algorithm was chosen as the same size of the corresponding training window,
e.g. the gap size for a 20-day training window was 20 days and so on. As such, in every scenario, the
entire data of 2013 were used step by step to test the performance of the algorithms as follows: at the
first step of each scenario, the gap began from 1 Jan  2013, while its corresponding training window
was the same size but came from the preceding period. For instance, for a 30-day gap, the first step
included training an algorithm based on the data of Dec 2012 and the testing period of the first month
of 2013. In the second step, the data of the first month of 2013 were used for training, while the data
of the second month of 2013 was considered as a gap, and this went to the end of 2013 consecutively.
As such, for the last step, the training window was the second last 30 days of 2013, and its
corresponding gap was the last 30 days of 2013. The only exception of the mentioned training strategy
was FBP as it needed a training dataset with at least a year to be developed. Therefore, here, the
training data for each gap was all data prior to that gap since the beginning of 2011. Overall, 18
variables, nine window lengths and eight gap-filling methods across five flux towers resulted in 6480
computations.

### 2.4. *Statistical performance measures*

Different statistical metrics were used to evaluate the performance of algorithms and enable
comparison between measured values from the flux towers with each gap-filling algorithm prediction.
These metrics included the coefficient of determination (R-squared) to measure the square of the
coefficient of multiple correlations (Devore, 1991), the variance of measured and modelled values ($S^2$)
to indicate how well algorithms could follow the variations of the recorded data, the root mean square
error (RMSE), the mean bias error (MBE) to capture distribution and bias of residuals, variance ratio
(VR) to compare the variance of estimated values with those of measured, and the Index of Agreement
to compare the sum of the squared error to the potential error (Bennett et al., 2013). Abbreviations and
formulas of these metrics are illustrated as follows (Bennett et al., 2013):





$$R^2 = \frac{[\sum(p_i - \bar{p})(o_i - \bar{o})]^2}{\sum(p_i - \bar{p})^2 \sum(o_i - \bar{o})^2} \tag{7}$$


$$S^2 = \frac{\sum(x_i - \bar{x})}{N - 1} \tag{8}$$


$$RMSE = \sqrt{\frac{\sum(p_i - o_i)^2}{N - 1}} \tag{9}$$



$$MBE = \frac{\sum o_i - p_i}{N - 1} \tag{10}$$


$$VR = \frac{\sigma_p^2}{\sigma_o^2} \tag{11}$$


$$IoAd = 1 - \frac{\sum_{i=1}^{n}(o_i - p_i)^2}{\sum_{i=1}^{n}(|p_i - \bar{o}| + ||o_i - \bar{o}|)^2} \tag{12}$$


where $o_i$ and $p_i$ are individual measured and predicted values respectively, $\bar{o}$ and $\bar{p}$ are the means of
o and p, and $\sigma^2$ is the variance. $S^2$ is calculated separately for the observed and predicted values with
the respective values defined as x that represents every observed or predicted value. All of these
metrics were calculated for each of the gap scenarios, and then the results of different windows were
concatenated. Afterwards, the yearly metrics were calculated to avoid Simpson's paradox or any
relevant averaging issue as described by (Kock and Gaskins, 2016). Moreover, the average of daily
and seasonal differences between the estimated and measured values, as well as the associated
variance were calculated and plotted.

## 3.    Results


### 3.1. *Fluxes*

*3.1.1 Fc*
Even though factors such as Fg and Fn are fluxes, we dealt with them as environmental drivers
since they drive the three major fluxes. The metrics used to evaluate the performance of the algorithms
(RMSE, $R^2$, MBE, IoAd and VR) (Table 4) illustrated that overall, the performance of these algorithms,
particularly the ML ones, was similar. The algorithms, however, showed different levels of sensitivity
to training/testing window length, e.g. the ANNs showed less sensitivity, whereas the FBP showed



the most sensitivity (Figure 1). The XGB provided the lowest values of RMSE and one of the highest
$R^2$, while the FBP and ELN had the lowest and highest values of RMSE and $R^2$, respectively.
*Table 4. The average amounts of performance metrics for each gap-filling algorithm regarding Fc, which includes all window*
*lengths and sites, ranked by RMSE using the Tukey's HSD test at the level of 5 per cent.*

| Algorithm | Mean RMSE | Mean $R^2$ | Mean MBE | Mean IoAd | Mean VR |
|-----------|-----------|-----------|----------|-----------|---------|
| XGB  | 3.53 [a]   | 0.56 | -0.44 | 0.89 | 0.59 |
| RF   | 3.56 [a]   | 0.54 | -0.38 | 0.90 | 0.70 |
| ANNs | 3.57 [a]   | 0.52 | -0.34 | 0.89 | 0.68 |
| SVR  | 3.81 [b]   | 0.47 | -0.33 | 0.86 | 0.79 |
| PD   | 3.89 [b]   | 0.45 | -0.36 | 0.80 | 0.53 |
| CLR  | 3.92 [b,c] | 0.46 | -0.37 | 0.80 | 0.54 |
| ELN  | 4.01 [c]   | 0.40 | -0.38 | 0.72 | 0.37 |
| FBP  | 4.15 [d]   | 0.44 | -0.06 | 0.77 | 0.68 |


These outcomes were expected for the XGB as it uses a more regularised model formalisation to
control over-fitting (Chen and Guestrin, 2016) that leads to better performance. The relatively poor
performance of FBP was also foreseen for unlike other algorithms, FBP did not use any feature to
estimate flux values, other than the previous time series of flux values. However, the weaker
performance of the ELN compared to CLR was unforeseen due to by adding two penalty components
to the regressed line, and the ELN is supposed to improve the long term prediction compared to the
traditional linear regression methods. Tukey's HSD (honestly significant difference) test at the level
of five per cent was applied to the results to find out whether the difference amongst the algorithms
was significant (Table 4). Where the null hypothesis was there is no significant difference between the
mean values of the RMSE. According to the results, there were significant differences between certain
algorithms, and the XGB, RF and ANNs were different from the rest, showing that these three
performed considerably better. Tukey's HSD test, however, did not reject the second error probability
between RF, XGB and ANNs meaning that the three algorithms were not significantly different from
each other. This result agrees with the results of (Falge et al., 2001) and (Moffat et al., 2007) in the sense
that ANNs are one of the best available algorithms, and there is no significant difference amongst the
appropriate algorithms. Nonetheless, it is worth mentioning that Tukey's HSD is well known as a
conservative test. That being said, despite no meaningful difference based on Tukey's HSD, XGB and
RF might have performed better than ANN, as the superiority of RF in gap-filling of methane flux has
recently been claimed by (Kim et al., 2020).





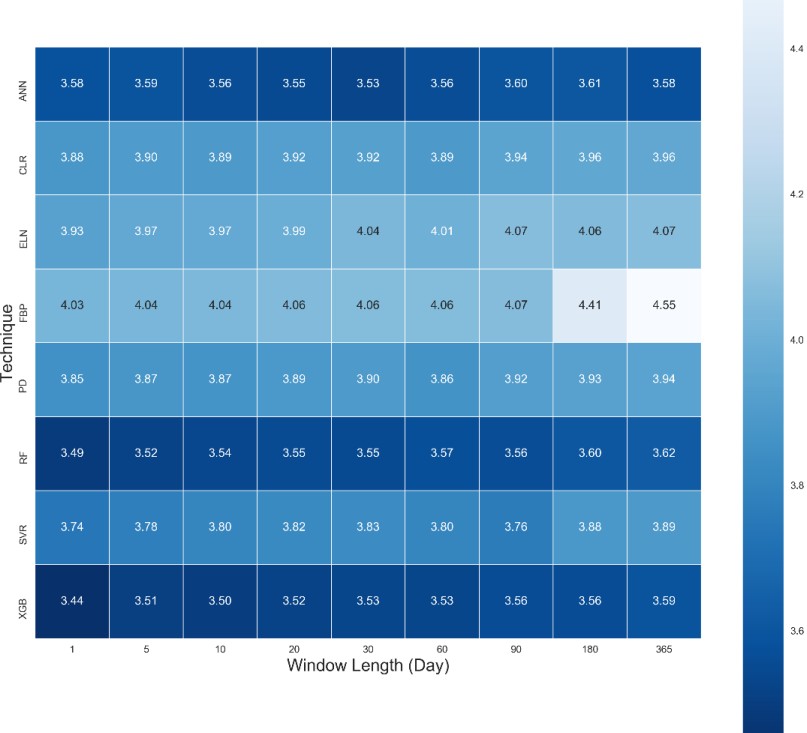


*Figure 1. A heat map of mean RMSE values of Fc across all sites based on 8 algorithms and 9 window lengths in 2013.*


To address the first objectives of this paper, finding out the sensitivity of each algorithm to the
training and testing window length, we used the averaged RMSE, $R^2$ and MBE for each window length
and gap size, using the output of all algorithms for all sites (Table 5). The outcome illustrates that the
longer the window length got, the bigger the amounts of RMSE became. Yet, no such pattern was
recognisable for the $R^2$ and MBE, particularly for the window lengths equal to or shorter than 60 days.
As a result, based on our scenarios (using the same length for training window and gap size), choosing
any training windows longer than 60 days, i.e. 90, 180 and 365 days, made the performance of the





algorithms worst. The phenomenon can be justified by the idea that longer windows do not let the
algorithms to accommodate seasonal changes and therefore, different physiological behaviour of
the canopy.

*Table 5. The average amounts of RMSE, $R^2$, and MBE for Fc gap-filling based on the window length including the outcome of all sites; the differences of RMSE values were tested using the Tukey's HSD test at the level of 5 per cent.*

| Window length | Mean RMSE | Mean $R^2$ | Mean MBE |
|:---:|:---:|:---:|:---:|
| 1-day | 3.74 [a] | 0.49 | -0.26 |
| 5-days | 3.77 [a] | 0.49 | -0.31 |
| 10-days | 3.77 [a] | 0.49 | -0.29 |
| 20-days | 3.79 [a] | 0.48 | -0.31 |
| 30-days | 3.80 [a] | 0.48 | -0.31 |
| 60-days | 3.79 [a] | 0.48 | -0.35 |
| 90-days | 3.81 [a] | 0.48 | -0.39 |
| 180-days | 3.88 [a] | 0.47 | -0.41 |
| 365-days | 3.90 [a] | 0.46 | -0.37 |

Besides, the metrics of the top three algorithms, XGB, RF and ANNs, did not show any sizeable
difference for the window lengths shorter than 60 days. As such, finding the ideal window length, at
least at this stage, is not distinctly noticeable and should rely on the local knowledge of each specific
site. Nevertheless, as mentioned earlier, the ideal window generally cannot be longer than 60 days,
unless for a monotonic ecosystem without a dramatic change during the year. According to the MBE
values, mainly, all algorithms had negative amounts of MBE, showing overestimation of the Fc values.
This bias varied from tower to tower and depended on the window lengths. For instance, absolute
amounts of the MBE were bigger in Gingin and Tumbarumba, while considerably smaller (closer to
zero) at AliceSprings Mulga and Calperum (results not shown). The lower leaf area index of the two
later sites, and thus their smaller amounts of photosynthesis is likely to be the reason that justifies the
outcome. FBP, nonetheless, provided substantially lower mean bias (-0.06) compared to the other
algorithms, which varied between -0.33 and -0.44.

Observations from the EC technique often include extremely low or high values, especially at
night, when some of the theoretical assumptions might be violated. The nature of the EC technique
associated with its practical challenges, often makes it difficult to distinguish between the good data
and the noise (Aubinet et al., 2012; Burba and Anderson, 2010). This problem seems to affect the
outcomes of the gap-filling algorithms in this research, as none of them performed ideally in capturing
the observed variance (Figure 2). Even though RMSE, $R^2$ and IoAd showed the superiority of the XGB,
RF and ANNs, the variance ratio between the estimated and measured values revealed different
information (Table 4), which is also recognisable in Figure 2. The variance ratios (VR) showed that
SVR captured the extreme values of Fc better than the other algorithms, 0.79 on average. The XGB, on





the other hand, provided smaller VR (0.59) compared with those of the RF (0.70) and ANNs (0.68),
especially for the window lengths longer than 10 days (not shown).

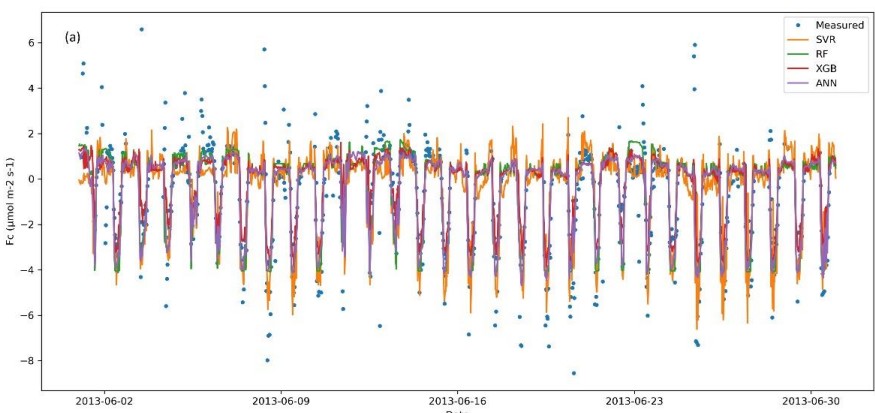

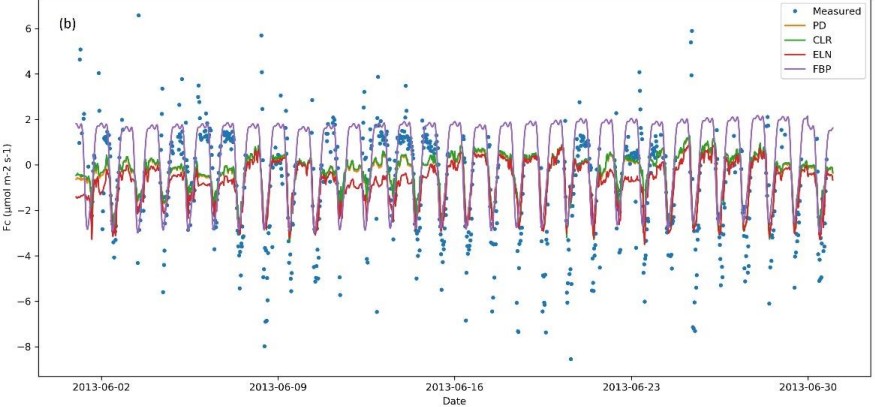


*Figure 2. Measured vs estimated values of Fc for Calperum based on the 30-day window during June (Austral winter) 2013*
This substantial smaller VR calls into question the ability of XGB to provide a solid gap-filling for long
gaps. The linear algorithms, CLR, PD, and ELN, performed worse with respect to the VR compared
to the ML algorithms. The estimated versus measured values of Fc for Calperum during June 2013
(Figure 2) confirms the information achieved by the VR. Based on the figure, and VR of 0.79, the SVR
captured the extreme values of Fc the best, whereas the ELN, as expected, performed the worst (0.37).
Although the XGB (VR of 0.59) provided relatively well while estimating the maximum values
(respiration), it was not capable of assessing the minimum values, thereby provided a constant
overestimation of NEE during the day. The RF (VR of 0.70), in contrast, captured both negative and
positive extremes better than the XGB, while the performance of the ANNs (VR of 0.68) was





somewhere in between. The rest of the algorithms performed poorly, particularly during the night,
except the FBP. It is noteworthy that CLR, PD, and ELN frequently predicted nocturnal
photosynthesis.
Apart from the objectives of this paper, tracing the performance of gap-filling algorithms
based on the hourly time step and seasonality has been as of the research interests. Thus, as an aside,
the differences between the average of estimations and measured values, as well as the difference
between the variances for the top three algorithms (XGB, RF and ANNs) were calculated for the 24-h
and seasonal ranges. These algorithms showed different anomalies in different towers and hours of
the day, except for Tumbarumba, where the patterns of anomalies were almost similar (Figure 3). The
average variance of differences was slightly lower during the night while the largest values of
anomalies usually occurred around noon, as expected due to the bigger variations of carbon uptake
caused by photosynthesis. Here, the RF and XGB performed better than ANNs, with the curves closer
to the basis, except for Tumbarumba. According to the seasonal anomalies, however, the algorithms
showed more similarities and closer outcomes, particularly for Tumbarumba, where all three
overestimated Fc values for the whole year (Figure 4). Similar to the 24-hour scale, the anomaly values
varied from site to site based on the season and the algorithm. Although the performance of the
algorithms was less here as against the daily scale, it seems that the XGB and RF still show superiority
over the ANNs. Apart from Tumbarumba, XGB, RF and ANNs showed a significant bias during
spring (July, August and September) in Howard Springs, when the site receives lower precipitation
due to the dry season.












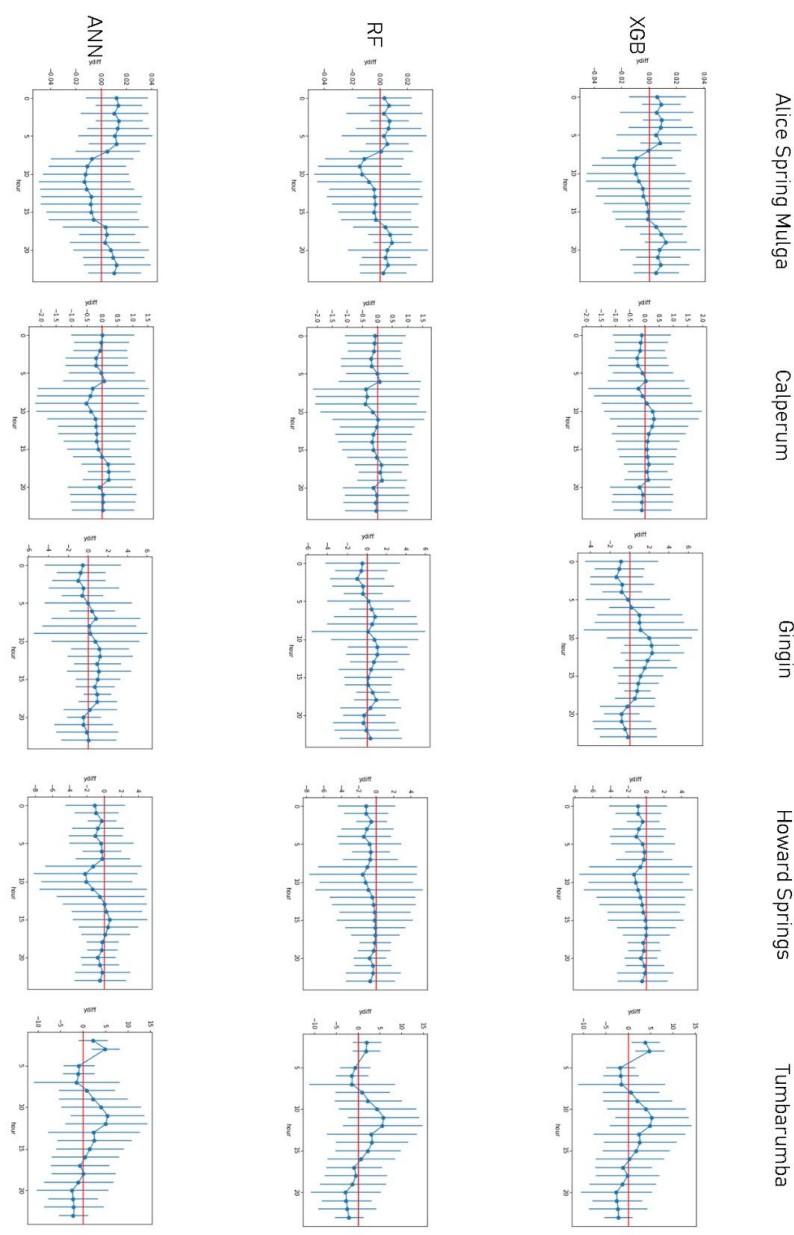


*Figure 3. 24-h Anomalies of XGB, RF and ANNs based on the Fc average of differences, and associated variances between the*
*estimated and measured values for all towers during 2013.*



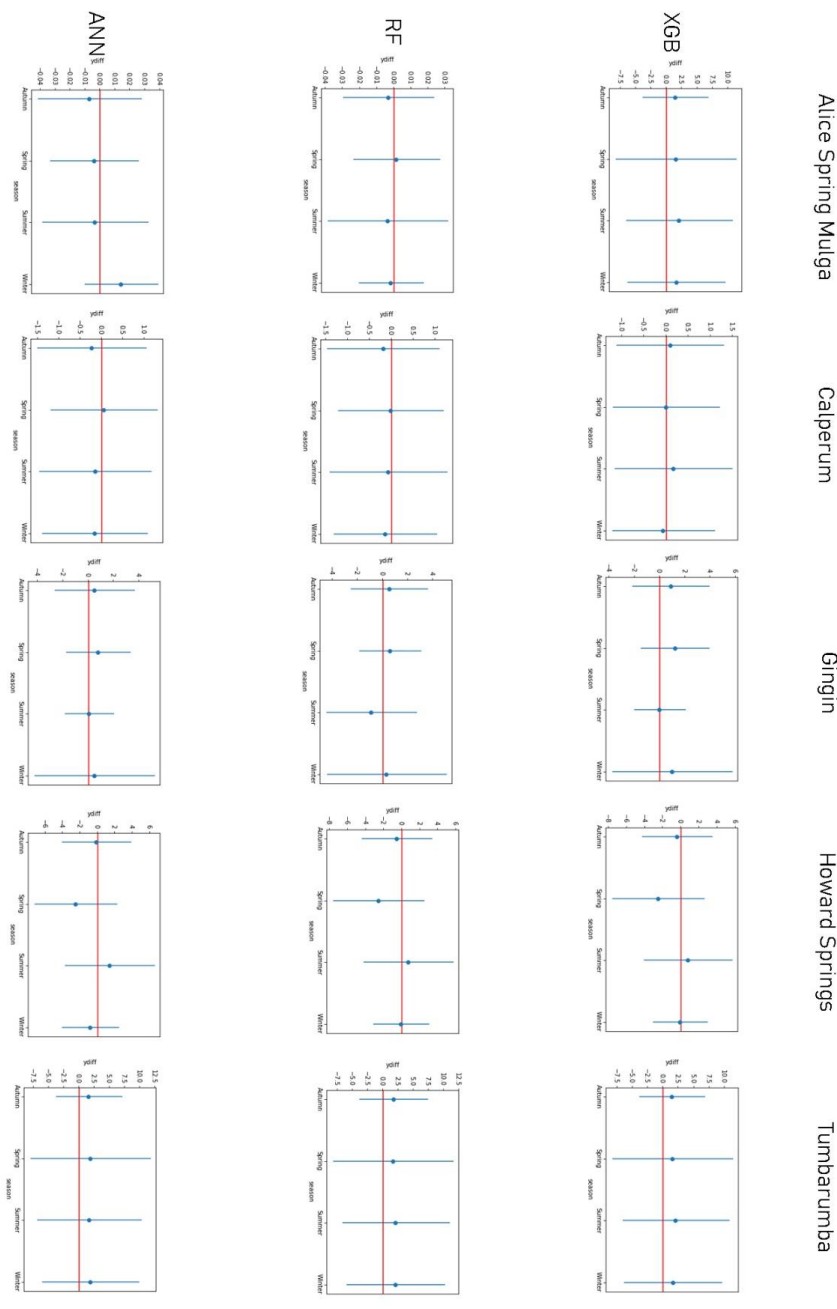

Figure 4. Seasonal anomalies of XGB, RF and ANNs based on the Fc averages of differences, and the associated variances between the estimated and measured values for all towers during 2013 (Jan, Feb and Mar as Summer, Apr, May and Jun as Autumn, Jul, Aug and Sep as winter).





### 3.1.2 Fe

The performance of algorithms for Fe was similar to that for Fc regarding RMSE, MBE and R2,
as shown in *Table 6*. This similarity was not surprising since these processes are partially coupled via
stomatal conductance (Scanlon and Kustas, 2010; Scanlon and Sahu, 2008). Again, the top three ML
algorithms performed better, with a significant superiority of the XGB and RF, as shown by the
Tukey's HSD (*Table 6*). Besides, the null hypothesis was not rejected while comparing FBP and SVR,
whereas the better performance of the other algorithms was confirmed. As a result, the FBP and SVR
provided the most unsatisfactory results in estimating Fe, according to the average values of the
RMSE. No significant improvement in RMSE occurred when the window lengths of training and
testing became shorter than 90 days, meaning that the performance of the algorithms did not vary
considerably from a 60-day to a one-day window. The results of CLR and PD were very similar to
those for Fc, showed lower RMSE and higher $R^2$ values as against ELN, but the ELN led to slight lower
MBE. The MBE values also showed moderately high values for the SVR, meaning that there was an
absolute bias in its outcome, which might be related to overfitting. The source of the bias shown by
the SVR algorithm (Figure 5), was because it could not capture the minimum values appropriately,
resulting in a considerable overestimation. A common issue in estimating Fe values, which had
affected all algorithms other than the FBP, was not assessing the negative values. In contrast to Fc
results, the ANNs did not perform as solid as the XGB and RF, which could be due to not being able
to capture the maximum values as satisfying as its rivals were.
*Table 6. The average of metrics for Fe gap-filling based on the algorithms, ranked by RMSE using the Tukey's HSD test at the*
*level of 5 per cent.*

| Algorithm (Fe) | Mean RMSE | Mean R² | Mean MBE |
|---|---|---|---|
| XGB | 37.27 [a] | 0.69 | -3.19 |
| RF | 37.98 [a] | 0.68 | -3.00 |
| ANNs | 40.62 [b] | 0.61 | -3.48 |
| PD | 42.45 [b,c] | 0.58 | -5.50 |
| CLR | 42.67 [b,c] | 0.58 | -5.95 |
| Eln | 43.48 [c] | 0.53 | -5.00 |
| SVR | 48.42 [d] | 0.53 | -21.08 |
| FBP | 49.46 [d] | 0.44 | 2.03 |







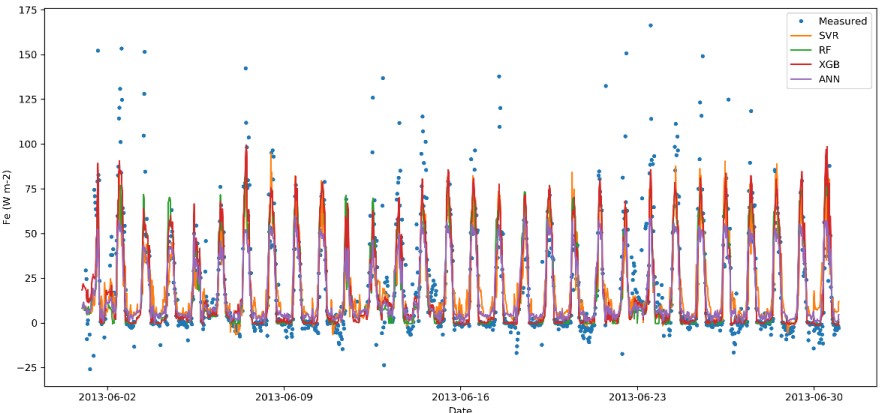


*Figure 5. Measured vs estimated values of Fe for Calperum based on a 60-day window during June 2013*

### 3.1.3 Fh

As with the other flux results, the metrics (RMSE, $R^2$ and MBE) showed slight superiority of
XGB and RF, as well as the inferiority of the SVR and FBP over the other algorithms (Table 7).
Likewise, the SVR provided relatively large negative values of MBE, showing considerable
overestimation. The Tukey's HSD test of the average RMSE values confirmed that the performance of
the FBP was significantly different from the rest at the level of 5 per cent, making FBP the weakest
performer for Fh. On the other hand, even though there was no significant difference amongst the
XGB, ANNs and RF, the first one was considerably superior over the other five algorithms as regards
the Tukey's HSD test. Like Fe, estimated values of Fh using SVR had a negative bias (Figure 6) because
it was not able to provide appropriate estimations of Fh negative values. In contrast, the ANNs
performed the best in capturing the minimum values, while the XGB and RF performed relatively
well, close to each other. Despite this superiority in assessing the minimum amounts, ANNs did not
carry out as solid as XGB and RF concerning the overall values, resulted in higher RMSE. Finally,
similar to the other fluxes, the PD performed slightly better than the CLR and ELN.
*Table 7. The average metrics for Fh gap-filling based on the algorithms, ranked by RMSE using the Tukey's HSD test at the level*
*of 5 per cent.*

| Algorithm (Fh) | Mean RMSE | Mean R² | Mean MBE |
|---|---|---|---|
| XGB | 37.26 [a] | 0.93 | -1.00 |
| RF | 38.08 [a,b] | 0.93 | -1.35 |
| ANNs | 40.48 [a,b,c] | 0.92 | -0.41 |
| PD | 41.83 [b,c] | 0.92 | -0.27 |
| CLR | 42.14 [b,c] | 0.92 | -0.05 |
| Eln | 42.28 [b,c] | 0.92 | 0.04 |
| SVR | 43.98 [c] | 0.91 | -8.28 |
| FBP | 67.19 [d] | 0.74 | 1.25 |

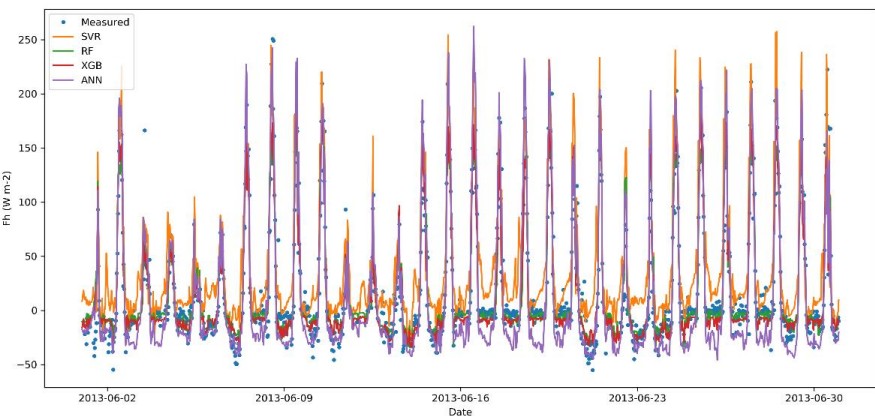


*Figure 6. Measured vs estimated values of Fh for Calperum based on a 60-day window during June 2013*

### 544    *3.2.  Meteorological and Environmental Drivers*

Since meteorological and environmental drivers are needed to fill the gaps of the three
substantial fluxes, Fc, Fe and Fh, the eight algorithms were used to fill the gaps of these drivers. The
metrics of $R^2$, RMSE, and MBE were calculated for all five towers and nine window lengths (16
meteorological and environmental drivers and three fluxes). Overall, for most meteorological drivers,
the linear algorithms, especially the CLR and PD, performed slightly better than the ML algorithms
such as the XGB, RF, ANNs and SVR, except for Ah, Fg and Fn. This unexpected superiority can be
explained based on the two following reasons. Firstly, unlike the fluxes, the input and output features
were the same here, e.g. Ta for Ta, which led to strong correlations (e.g. up to 0.99 for atmospheric
pressure - ps) as well as strong linear relationships between the independent and dependent features.
These strong correlations helped the linear algorithms to perform well, while nullified the ability of
ML algorithms to capture non-linear behaviour of complicated problems. Second, the slight inferiority
of ML algorithms could be due to data noise where simple linear algorithms such as the CLR are
usually less sensitive to the noise relatively. Therefore, over-fitting is not an issue for them when the
number of observations is big enough (i.e. at least 10 to 20 observations per parameter (Harrell, 2014)).
The exceptions were Ah, Fn and Fg, for which values were estimated more accurately by the XGB,
ANNs and RF, especially the latest one (the RMSE of 30.23 versus 35.24 provided by the RF and CLR
for Fg, respectively). Tukey's HSD test for the mean RMSE values of Fg confirmed that The XGB,
ANNs and RF provided better results at the level of 5 per cent, while, like all other fluxes and drivers,
the FBP confirmed to be the worst algorithm (Table 8). Yet, according to the same test for the other
drivers, there was not any significant difference between the algorithms, other than the FBP, which
provided the most significant mean values of the RMSE (results not shown). Importantly, though,
none of the algorithms offered adequate estimations for soil moisture (Sws), particularly in drier
regions. This weak performance happened because Sws changes dramatically during rainfall in a



pulsed manner often from zero to saturation in short space of time, whereas, the algorithms had been
trained based on the datasets mostly reflecting non-rainy periods. These datasets, consequently, could
not fit the algorithms in a way that they could estimate Sws accurately when precipitation occurs and
the soil moisture increases dramatically. For instance, in a  wet region like Tumbarumba, where the
soil faces rainy days frequently, the time series are much less spikey. Thus, the overall performance
was better in these regions compared with the drier ones, e.g. $R^2$ of 0.43 and 0.25 on average for
Tumbarumba and Calperum, respectively. Besides, the dataset used to gap-fill the soil moisture was
a model derivation from gridded data or regional reanalysis and therefore, can be not close to reality.
Another challenge of estimating soil moisture comes from the low spatial coherence of soil moisture
is that it can be extremely different just a couple of hundred metres away, due to storms, topography,
soil structure heterogeneity, etc. (Reichle et al., 2004; Sahoo et al., 2008).

*Table 8. The average amounts of RMSE for Fg gap-filling based on the algorithms, using the Tukey's HSD test at the level of 5*
*per cent.*

| Algorithm (Fg) | Mean RMSE |
|---|---|
| RF [a] | 30.17 |
| XGB [a, b] | **30.70** |
| ANNs [b, c] | 30.86 |
| SVR [c] | 32.77 |
| CLR [d] | 34.93 |
| PD [d] | 34.94 |
| ELN [d] | 34.94 |
| FBP [e] | 39.10 |


## 4.   Discussion

All algorithms performed similarly in estimating the meteorological and environmental drivers
(turbulent fluxes included) across all stations, except the FBP, which performed poorly for it did not
use any ancillary data. The best results were achieved using training/gap windows of 60 days or
shorter, while the worst results obtained for the most extended window, 365 days. Although most of
the algorithms performed almost equally well in estimating of meteorological and environmental
drivers, the linear algorithms, the CLR, ELN and PD, performed slightly better (not significant using
a Tukey's HSD test, though). The only clear exception was Fg, which the RF provided more accurate
and robust estimations. The ML algorithms, on the other hand, showed their superiority over the
linear algorithms while estimating the main fluxes, Fc, Fe and Fh. For Fc, the XGB, RF and ANNs
performed significantly better than the SVR, FBP and all linear algorithms, i.e. the CLR, PD and ELN.
The superiority of the ML algorithms, as well as their close performance,  agreed with the results of
previous researches, e.g. (Falge et al., 2001; Moffat et al., 2007), that showed the superiority of non-
linear algorithms and no significant difference amongst the top algorithms in estimating Fc. Besides,
the slight superiorities of RF over ANNs, mainly unnoticeable by a conservative test like Tukey's HSD,
confirms RF performs better regarding the EC flux gap-filling (Kim et al., 2020).



The XGB was the most novel ML algorithm used in this research and based on the most
performance metrics provided comparatively robust results in estimating the fluxes. However, the
XGB failed to capture the minimum values of Fc as against the SVR and RF, and thus, provided biased
results, while assessing the maximum values of Fc well. In estimating the meteorological drivers, the
XGB did not show any superiority over the other algorithms, especially the linear ones. Hence, we do
not recommend the XGB as an alternative to the current alternative algorithms, especially for long
gaps. Nevertheless, because of its local superiorities, this algorithm is suitable to use in an ensemble
model alongside the algorithms with different weakness points.
The RF was the best all-around algorithm amongst the eight algorithms used in this study,
providing the best estimates of the fluxes (similar to XGB) but also captured both minimum and
maximum values of Fc. Unlike the RF, all other algorithms generally struggled with estimating either
minimum or maximum values of major fluxes, comparatively. It also provided the best results for Fg,
where the linear algorithms did not perform well. Another advantage of the RF over the XGB was that
it required less time (approximately a quarter) for training, which was a challenge while using the
XGB.
The ANNs estimated the fluxes better than the linear algorithms, notably for Fc, yet not as
robust as the XGB and RF in general. For Fc and Fh, the ANNs provided bias, mainly due to
overestimation of minimum values when the window lengths were 90 days or longer. However, since
the superiority of the XGB and RF was not considerable, it is difficult at this point to suggest using
XGB or RF as better alternatives. That is because ANNs have been checking out for a long time in
different locations and considered as one of the most reliable algorithms in the field for more than a
decade (Aubinet et al., 2012; Hagen et al., 2006; Kunwor et al., 2017; Moffat et al., 2007). Furthermore,
there are a wide variety of different ANNs algorithms used in the field (Beringer et al., 2016b; Hagen
et al., 2006; Isaac et al., 2017; Kunwor et al., 2017; Moffat et al., 2007), and this minor superiority of RF
and XGB cannot be generalised without enough additional proves. As such, we suggest other
researches to use the RF, especially regarding Fh and Fc alongside the ANNs to find out which one
performs better in the challenging scenarios, e.g. when the gaps are long. Another option is to develop
ensemble models using since, according to the literature, there is no room to improve the results
substantially based on a single algorithm (Moffat et al., 2007).
On the other hand, a model with a higher level of flexibility is required in the field (Hagen et
al., 2006; Kunwor et al., 2017; Richardson and Hollinger, 2007). Finally, according to the environmental
drivers, The ANNs, like the other ML algorithms, could not show a consistent superiority over the
linear algorithms. Therefore, we do not recommend using ML algorithms in such scenarios, except for
Fg, for which RF seems to be a better option.
The SVR showed consistent inferiority over the other ML algorithms and did not fulfilled our
expectations, neither for the meteorological drivers nor for the major fluxes. The only strength of the
SVR  was that it captured the extreme values better than any other algorithm, as revealed in the plots
of Fc. However, according to its larger RMSE amounts, the mentioned advantage seems to be achieved
suspiciously and might have occurred due to over-fitting. This dubious performance shows the SVR
is more vulnerable to the over-fitting issues regarding these types of data. Hence, we suggest the SVR





not to be used in any kind of environmental modelling related to the reviewed drivers and fluxes,
whatsoever.
The CLR, the simplest algorithm used in this research, provided a comparatively acceptable
performance in estimating the meteorological drivers, except for Fg. This algorithm, however, could
not perform well in assessing the fluxes, especially Fc, mainly because of its inability to capture the
extreme values caused by the non-linear nature of Fc. Overall, considering the CLR simplicity,
resource-saving and robust performance for drivers, this algorithm seems to be the most suitable way
to fill the gaps of meteorological parameters in similar scenarios, where the same ancillary dataset is
available.
The PD performed slightly better than the CLR, yet it could not fulfil the expectations to show
a significant superiority over the other linear algorithms used in the research. This unforeseen weak
performance can be explained due to a couple of reasons. First, one of the assumptions of using the
PD is that the behaviour of the cross-sections, here towers, is similarly under the similar conditions
(the independent variables), and the only thing leads to the difference is the specific characteristics of
each individual cross-section. Contrariwise, it seems that the five towers selected in this research
violated this assumption due to their absolute different ecosystems. Based on the previous studies in
which the PD performed satisfying (Izady et al., 2013, 2016; Mahabbati et al., 2017), (Izady et al., 2016)
and (Mahabbati et al., 2017), it appears that a decent level of homogeneity is vital for the PD to perform
satisfactorily. As in all previous cases, the ecosystem of the cross-sections had significant similarities,
and the distance between them were tens to hundreds of kilometres, not thousands. Therefore, the
characteristics of cross-sections, such as radiation, climate, rainfall, etc. had considerable more
similarity and homogeneity compared with the towers used in this research. Finally, it is worth
mentioning that PD has been commonly used to analyse the time series with a time resolution of
weekly or longer, with some exceptional daily-scale cases. In this research, the resolution of data was
half-hourly instead, which dramatically increased the computational demands of the algorithm, led
to days of processing for a single run. This demand happened because the algorithm creates a dummy
variable for each time step and the relevant matrix of variables becomes too large to compute by a
regular PC. Considering the expenses of this algorithm, we recommend other researches not to use
PD when the time resolution is shorter than daily. Despite the limitation, we still encourage further
using of PD whenever there is a decent level of homogeneity amongst the cross-sections and the time
resolution is daily or longer (ideally weekly or monthly).
The ELN, as a hybrid linear model, did not show any superiority over the CLR, despite its
modifications to provide more accurate estimations. Even though ELN performed well in estimating
the drivers with slight supremacy in some occasions, e.g. Fld, the CLR is a more proper algorithm to
choose for gap-filling the drivers due to its simplicity and less calculation requirement.
The FBP was a unique algorithm used in this research, as it did not use any independent
variables to estimate the values of drivers and fluxes.  The FBP performance was significantly more
unsatisfactory than the other algorithms. Therefore FBP cannot be considered as a reliable alternative
for current algorithms to fill the gaps, especially the long ones.



## 5. Conclusions

Eight different gap-filling algorithms for estimating 16 meteorological drivers as well as the three key ecosystem turbulent fluxes (sensible heat flux (Fh), latent heat flux (Fe), and net carbon flux (Fc)) were investigated and their performance evaluated based on the datasets of five towers in Australia. Overall, three ML algorithms, XGB, RF and ANNs, performed nearly equally well and significantly better than their linear rivals (the CLR, PD, and ELN) in estimating the flux values. However, the linear algorithms performed almost as equally well as the ML algorithms in assessing the meteorological drivers. Amongst these eight algorithms, the RF showed the highest level of robustness and reliability in estimating the Fc, as its closest rival, the XGB, could not capture the minimum values equally well, despite providing slightly better RMSE and $R^2$. The PD was expected to perform better than the linear methods and hoped to compete with the ML algorithms in estimating the fluxes, but it failed to do so. The SVR was the only ML algorithm that did not perform at the same level as the rest ML algorithms and was suspected of enduring over-fitting issues. Considering the outcomes of the other researches undertaken in the OzFlux Network, e.g. (Cleverly et al., 2013; Isaac et al., 2017), none of the ML algorithms used in this research was proven to provide substantially better flux estimations compared with the standard method (ANNs). Nonetheless, amongst the algorithms tested in this research, the RF showed some potential capabilities as an alternative due to its more consistent performance regarding the long gaps. Eventually, we recommend suggestions below to improve the results for similar prospective researches, as well as the QC and gap-filling procedure of OzFlux Network:

1) Since the RF remained more consistent compared to its competitors -including the ANNs-, It is a good idea to use RF alongside the commonly used algorithms in the challenging scenarios, such as long gaps, to figure out whether this superiority can be generalised.

2) It appears that, even after three levels of quality control process done by the PyFluxPro platform, the data are still noisy. This noisy data are an essential source of both uncertainty and inaccuracy of the outcome, regardless of the algorithm used to gap-fill the data. As a result, another level of quality control methods, such as Wavelets or Matrix Factorialisation, in addition to the current classical ones used by the PyFluxPro and other similar platforms, can probably improve the data quality and thereby improve the final imputation results.

3) For future researches, using recurrent neural networks (RNNs) instead of feedforward neural networks (FFNN) could improve the predictions. That is likely because RNNs help the model to consider temporal dynamic behaviour of time series, as unlike FFNN, wherein the activations flow only from the input layer to the output layer, RNNs also have neuron connections pointing backwards (Géron, 2019). This demand to an algorithm capable of considering time has been mentioned in previous researches as one of the reasons why testing the new algorithms is needed (Richardson and Hollinger, 2007).

3) Developing ensemble models using algorithms with different weaknesses and strengths may also enhance the results where a single algorithm shows performance deficiency.



4) Given that some of the environmental drivers affect the Fc differently during the day versus night,
separating the diurnal and nocturnal datasets to train the algorithms possibly entails an improvement
in the outcome. Mainly because of the $u^*$ threshold filtering and other problems associated with the
nocturnal period, the portion of diurnal data is generally, by far, outweighs the nocturnal data portion,
which potentially leads to a bias in the algorithm.
5) The same solution as number 4 is suggested for soil moisture estimation, as the behaviour of the
system on sunny days is utterly different from its conduct during the rainy periods. Moreover, the
system memory and the antecedent condition are undeniable features associated with soil moisture
(Ogle et al., 2015). Therefore, using the models that are capable of addressing these considerations are
more likely to improve the estimations.

## 6.    Data availability

The data were used in this research are available through the following sources: The L3 and L4
data are accessible from the OzFlux data portal (http://data.ozflux.org.au/portal). Current ACCESS-R
and data are available from the BoM OPeNDAP server (https://www.opendap.org/). Likewise, the
data coming from the BoM AWS are accessible from (http://www.bom.gov.au/climate/data). Lastly,
the    BIOS2    data    are    accessible    from    the    ECMWF    datasets    portal
(https://www.ecmwf.int/en/forecasts/datasets). All data used in this research are available in this
repository address: (https://research-repository.uwa.edu.au/en/datasets/a-comparison-of-gap-filling-
algorithms-for-eddy-covariance-fluxes); DOI: 10.26182/5f292ee80a0c0.

*Author contributions.* The ideas for this study originated in discussions with A. Mahabbati, J. Beringer,
and M. Leopold. A. Mahabbati carried out the analysis, supported by I. McHugh and P. Isaac. The
paper was prepared with contributions from all authors.
*Competing interests.* The authors declare that they have no conflict of interest.
*Acknowledgements.* The authors would like to acknowledge Terrestrial Ecosystems Research Network
(TERN) (www.tern.gov.au) and the OzFlux Network as a part of TERN for supporting the grants and
providing the required data, respectively. A. Mahabbati also personally thanks Prajwal Kalfe, Caroline
Johnson and Cacilia Ewenz for their support as regards Python programming, English academic
writing and PyFluxPro technical issues.

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
