# Peer review of "A comparison of gap-filling algorithms for eddy covariance 2 fluxes and their drivers"

_Geoscientific Instrumentation, Methods and Data Systems, 2020_

## Referee Comment (RC1) · Thomas Wutzler (Referee) · 2 Oct 2020

General comments

The paper by Mahabbati et al. presents an updated comparison of gap-filling algorithm, which are an important tool in the analysis of data from eddy-covariance sensors and understanding the ecosystem functioning. Their methodology is oriented at the Australian version of the data processing chain taking into account information in addition to the eddy-stations from weather forecasting models and from BIOS2 model data integration environment. For gap-filling of meteorological drivers, they corroborate previous findings of complex methods being not much better than simple methods. Contrary, for the carbon fluxes itself they find a better performance of the machine learning (ML)

based approaches.

This study is a valuable contribution to the Australian setup. However, their findings are difficult to transfer to other processing setups and other sites. Hence, the paper is a quite special application and in the current form better suited for an Australian journal. I encourage the authors for a major revisions to extent their study to setups that are also applicable at other sites for submission to GI.

I have several major concerns, which I state here and explain below. First, I propose to add a comparison with fitting the models to only data that are commonly available at other sites. Second, the methodology needs to be updated to introduce gaps at random positions in time instead of all starting at 1st of January to avoid confounding of gap-length with seasonality. Third, I propose to include the MDS algorithm that was simple but well performing at previous gap-filling comparisons and a "business as usual" for gap-filling NEE at many sites.

Specific comments

In order to be usable at other sites, the methods should be compared in addition to the presented setup by using only data commonly available at eddy-covariance sites, which are the measurements themselves (Fc, Fh, Fe) together with ancillary measurements (Rg, VPD, rH, Tair, Tsoil, Ustar, precip, wind speed, and wind direction), and maybe another comparison using in addition more detailed radiation measurements and ground heat flux and soil water storage (Table 2).

In the current comparison setup, the larger gap-lengths comprise a larger proportion of other seasons, while the short gap-lengths only comprise summer records. Hence, the conclusions on gap-lengths are confounded with seasonality. I suggest to randomly distribute gaps in the portion of the entire data series with sufficiently high proportion of non-missing original data. Moreover, most data-processing setups will not fit a model for each gap tailored at the gap-length. Hence, I suggest to introduce several gaps (of a given length) across the entire dataset (say of proportions of 40% and 70% of the

data according to p6L215) and let each methods fill all these gaps and compute the statistics across all the gaps but also of the aggregated annual value. In this way a recommendation can be presented that is closer to the gap-filling as applied at many sites. The decision to adjust the training window to the gap-length is very difficult to compare to other gap-filling of real time series where gap-lengths vary. Most investigators will not effort to fit a model around each gap. I suggest training the methods on a shifting window and filling all gaps inside this window, and for efficiency use only few increasing window lengths of the training.

Moffat et al. (2007) concluded that the quite simple and widely applied MDS algorithm for filling Fc, i.e. NEE time series, which is using only the common variables NEE, Rg, Tair, and VPD as predictors. What are the reasons to omit this for many sites "business as usual"-algorithm? The computation can even be outsourced to the online-tool provided by the MPI-BGC Jena.

P7L226: Were all the eight drivers used or a subset of them, maybe different by method? What is q? The formulation "by trial and error" needs more explanation.

P10L308: Here it does not become clear what cross-sections have been used. I imaged some categories based on similar environmental conditions or day/night time. This only becomes clear in the discussion, in that data from other sites have been used with site as cross-section. This cross-site gap-filling is hard to transfer to other studies. In what respect does the PD model differ from a classical mixed effects models?

P27L720 Conclusions 4 and 5 are mere speculations given the results presented in the paper. They should be moved to the discussion. Contrary to the suggestion 4, I hypothesize that using net radiation as a predictor should handle this case already well (at least with RF). Otherwise, I suggest first trying to add a nighttime/daytime flag to the set of predictors before splitting the dataset.

P1L35: Currently, I was confused reading the abstract. It was hard for me to spot the distinction between filling of environmental drivers and filling of fluxes. This can be

formulated more clearly.

Technical corrections

P2L43: This formulation does not become clear to me.

Tab 2: I suggest indicating the commonly used abbreviation for the fluxes in parentheses in addition to the notation of the paper (NEE, LE, H).

P11L331: typo: "non-periodic"

eq 12: one bar too much.

P23L583: I suggest to provide another table with method abbreviations or repeat the abbreviations at the beginning of the discussion. By this way you do not force your readers to study the methods section first.

---

## Referee Comment (RC2) · Anonymous Referee #2 · 20 Oct 2020

GENERAL COMMENTS

The paper presents a detailed evaluation of eight algorithms for gap-filling time series data, using eddy covariance data as a target for the comparisons. The content about the algorithms and the metrics for comparisons are a strong feature of the paper. However, it is more limited in advancing the knowledge of best practices for eddy covariance and micro-meteorological data gap-filling. In other words, the evaluation of the algorithms against each other is of interest, but the chosen test domain is not clearly impacted.

It seems that to really benefit the knowledge of methods for gap-filling eddy covariance data, longer time series and more representative gap scenarios would be necessary, as well as a clear comparison to more established methods. Multi-year datasets are

key to properly evaluate these algorithms. Such datasets are now widely available, so it is unclear why only 2013 was used. With this aspect in mind, it seems clear that in the evaluation of the first objective of the paper longer gaps led to disproportional increases in uncertainty. This might not have happened if other years without gaps for the same season were available, for instance. More direct comparisons to "classic" gap-filling algorithms would have helped in this evaluation. Implementations of algorithms such as MDS are now widely available, including as part of OZFlux's own OzFluxQC software package. The comparison of newer methods is informative, but unless compared to currently used solutions, it's hard to assess the improvement. Although the authors are correct, and performance of the MDS algorithm was shown to be comparable to ANNs before, parameterizing MDS is much simpler (no choices in layers, nodes, iterations, or window sizes) and would lead to a more robust and clear comparison.

Should the authors choose to really focus on the comparison among the methods presented, I would suggest adding all the comparison metrics RMSE, R2, MBE, etc., for all sites individually and combinations thereof as supplementary materials, making this a valuable and thorough comparison of methods, and reducing the focus from the application to eddy covariance. If the intention really is to show the impact on EC, longer time series and more direct comparisons to current methods would be necessary.

SPECIFIC COMMENTS

On the ancillary datasets, it seems they introduce some entanglement to this evaluation. One of the key advantages of purely empirical methods, such as the ones presented in the paper, is that they will not be biased by predefined models (like the reanalysis datasets) or atmospheric interferences (like the MODIS data). After an evaluation without datasets such as these, adding them to improve the methods would be a natural choice. However, without the unbiased evaluation it is hard to qualify the sources of uncertainty in the paper's evaluation.

Although the performance criteria selected for the paper work well, it is curious to see

that the methods all seem to represent high variabilities but fail to capture the extremes, as the authors point out for CO2 and latent heat fluxes – and this doesn't seem to be the case for sensible heat flux. Could this be an issue of the underlying data requiring further quality control before the gap-filling methods are applied? Or maybe this is an artifact of the period selected in the examples?

The following claim requires either more details or a reference, otherwise it's not possible to know what concerns/challenges the authors are referring to and what aspects of gap-filling the paper is aiming to address: "...there are some serious concerns regarding the challenges associated with the technique, e.g. data gaps and uncertainties."

The +/-25gCm-2y-2 (Moffat et al. 2007) and +/-30gCm-2y-2 (Richardson & Hollinger 2007) are dependent on the underlying datasets used for the evaluation. These numbers should not be taken as general benchmarks.

In the sentence "Nevertheless, one of the concerns regarding this algorithm is that the independent variables, here meteorological drivers, might be auto-correlated." it is unclear why this would be a concern, since the meteorological drivers being auto-correlated is one of the assumptions that allow the MDS method to work.

The sentence "This challenge becomes acute when the gaps happen within a period when the ecosystem behaviour is changing and thereby showing different response under similar meteorological conditions." is another reason why multi-year datasets should be used to compare these algorithms.

The gap scenarios and training windows selected are highly structured and rigid. It's unclear how the evaluation over these scenarios would translate into real-world examples, which have both structured gaps (e.g., from sensor failures) and arbitrary gaps (e.g., from data filtering). It seems it would be important to use are least one scenario with gaps and training data both randomized, and also combinations of lengths for gap windows and training windows.

TECHNICAL CORRECTIONS

— Abstract

- The acronyms RF and CLR were referenced before being defined

- "...RF provided more consistent results with less bias, relatively." It would be clearer to describe "relatively" to what in this sentence.

- This sentence is a bit unclear "In each scenario, the gaps covered the data for the entirety of 2013 by consecutively repeating them, where, in each step, values were modelled by using earlier window data." Were measured and modeled data used simultaneously in training?

— Introduction

- "...and not measured at the point." Maybe could be "not measured at a point scale"?

- A more classic reference for FLUXNET is: Baldocchi et al. 2001. FLUXNET: A New Tool to Study the Temporal and Spatial Variability of Ecosystem-Scale Carbon Dioxide, Water Vapor, and Energy Flux Densities. BAMS, 11: 2415-2434.

- And more appropriate references for EUROFLUX and AmeriFlux are: Aubinet, M. et al. 1999. Estimates of the Annual Net Carbon and Water Exchange of Forests: The EUROFLUX Methodology. Advances in Ecological Research, pp. 113–175. Law, B. 2007. AmeriFlux Network aids global synthesis. Eos, 88, 286–286. Novick, K. A. et al 2018. The AmeriFlux network: A coalition of the willing. AFM, 249:444-456.

- "Despite the capability of EC to frequently validate process modelling analyses..." might be more precisely phrased as something like "Despite EC data being frequently used to validate process modelling analyses..."

- "[...] Moffat et al. (2007) compared a couple of different commonly-used gap-filling algorithms"; in fact, Moffat et al. 2007 compared 15 gap-filling techniques.

— Materials and Methods

- "and Tumbarumba form 2011 to 2013..." form -> from

- "Each algorithm was tuned up individually using gird search,..." gird -> grid

— Results

- Even with a maximum zoom in the PDF file, it is rather hard to read the axis for Figures 3 and 4

— Discussion

- This sentence is unclear: "That is because ANNs have been checking out for a long time in different locations and considered as one of the most reliable algorithms in the field for more than a decade"

---

## Author Comment (AC1) · 10 Dec 2020

General Comments:

RC01: The paper by Mahabbati et al. presents an updated comparison of gap-filling algorithm, which are an important tool in the analysis of data from eddy-covariance sensors and understanding the ecosystem functioning. Their methodology is oriented at the Australian version of the data processing chain taking into account information in addition to the eddy-stations from weather forecasting models and from BIOS2 model data integration environment. For gap-filling of meteorological drivers, they corroborate previous findings of complex methods being not much better than simple methods. Contrary, for the carbon fluxes itself they find a better performance of the machine

learning (ML) based approaches. This study is a valuable contribution to the Australian setup. However, their findings are difficult to transfer to other processing setups and other sites. Hence, the paper is a quite special application and in the current form better suited for an Australian journal.

AC: Even though the data used in this paper came from Australia, the focus was to find out whether ML algorithms other than ANNs can provide more robust results regarding gap-filling of drivers and fluxes. That being said, the towers are selected just as samples to compare the performance of different algorithms. In that sense, the paper is algorithm-oriented rather than Australian-style oriented, and the output is suitable for all members of the FLUXNET. Please note that the diversity amongst the towers has been wide, and there is less likely that an algorithm like RF, which has consistently provided a robust performance in all the five different climates and sites, perform poorly in other parts of the world, or with different input features. Keep in mind that the initiation of this study is to compare different algorithms.

RC01: I encourage the authors for a major revisions to extent their study to setups that are comment also applicable at other sites for submission to GI.

I have several major concerns, which I state here and explain below. First, I propose to add a comparison with fitting the models to only data that are commonly available at other sites. Second, the methodology needs to be updated to introduce gaps at random positions in time instead of all starting at 1st of January to avoid confounding of gap-length with seasonality. Third, I propose to include the MDS algorithm that was simple but well performing at previous gap-filling comparisons and a "business as usual" for gap-filling NEE at many sites.

AC: For the first suggestion, since the main goal of the study was to compare different gap-filling algorithms, we do not believe changing the input data leads to a difference in the relative performance of the algorithms. Moreover, as mentioned in the materials and methods, a variety of climates is involved in this study (Beringer et al. 2016), which

makes the results useful for different types of audiences. Please note that the area of Australia is almost as twice as big as the Western Europe, and it has a large variety of climates.

As for the second suggestion, we think we need to clarify the scenario by which we filled the gaps. Since in each gap-filling round the entire 2013 data have been covered by multiple steps for the gap windows shorter than 365 days. For instance, when the gap window is 30 days, the script does the training and testing process 12 times in a row so that it fills the entire 2013, where in each step the model is trained using the data of the previous month. We know that it might not be the best way to fill the gaps, but please note that this paper is the first one of a series of papers which the corresponding author has been working on for his PhD. He has been undertaking the second part of his research by superimposing the gaps randomly and he is writing it down as the second paper of his thesis. So the suggestion will be fulfilled in the second paper wherein the corresponding author has used the same data. However, if the reviewer insists on changing the gap-filling scenario, we would be happy to do the process again, and applying random gaps, instead.

For the third suggestion, including the MDS, we accept the suggestion and will do so.

Specific comments:

RC01: In order to be usable at other sites, the methods should be compared in addition to the presented setup by using only data commonly available at eddy-covariance sites, which are the measurements themselves (Fc, Fh, Fe) together with ancillary measurements (Rg, VPD, rH, Tair, Tsoil, Ustar, precip, wind speed, and wind direction), and maybe another comparison using in addition more detailed radiation measurements and ground heat flux and soil water storage (Table 2).

AC: We believe that it is a useful suggestion. However, as mentioned earlier, the main goal was to compare different gap-filling algorithms and it is less likely that changing the input features makes any change in the comparative performance of the models.

For instance, Kim (Kim et al 2019) compared ANNs, RF, SVR and MDS to fill the gaps of Methane flux with different input features than this study, and the performance ranking amongst the ML methods was quite similar to this paper: RF outperformed the rest, and ANNs outperformed SVR. Besides, the data used in that research came from North America.

RC01: In the current comparison setup, the larger gap-lengths comprise a larger proportion of other seasons, while the short gap-lengths only comprise summer records. Hence, the conclusions on gap-lengths are confounded with seasonality. I suggest to randomly distribute gaps in the portion of the entire data series with sufficiently high proportion of non-missing original data. Moreover, most data-processing setups will not fit a model for each gap tailored at the gap-length. Hence, I suggest to introduce several gaps (of a given length) across the entire dataset (say of proportions of 40% and 70% of the data according to p6L215) and let each methods fill all these gaps and compute the statistics across all the gaps but also of the aggregated annual value. In this way a recommendation can be presented that is closer to the gap-filling as applied at many sites. The decision to adjust the training window to the gap-length is very difficult to compare to other gap-filling of real time series where gap-lengths vary. Most investigators will not effort to fit a model around each gap. I suggest training the methods on a shifting window and filling all gaps inside this window, and for efficiency use only few increasing window lengths of the training.

AC: This is a good suggestion, and this is a better approach in general for a realistic gap-filling process. However, we want to point a few things out: First, It seems that the paper explanation about the gap-filling approach is not clear enough. we think we need to clarify the scenario by which we filled the gaps. Since in each gap-filling round the entire 2013 data have been covered by multiple steps for the gap windows shorter than 365 days. For instance, when the gap window is 30 days, the script does the training and testing process 12 times in a row so that it fills the entire period of 2013, where in each step the model is trained using the data of the previous month. We

confirm that this approach might not be the best way to fill the gaps, but please note that this paper is the first one of a series of papers which the corresponding author has been working on for his PhD thesis. The second paper of his research is been doing by superimposing the gaps randomly. So the suggestion will be fulfilled in the second paper anyway.

Second, the main goal of this study has been comparing the performance of different ML-based gap-filling algorithms. In that regard, it is less likely to be a considerable relative performance difference between the scenarios whereby the gap-fillings are carried out.

According the points above, I believe that changing the gap-filling scenarios would not make any significant change in the relative performance of the algorithms. Considering the main goal of the current study, which to compare different algorithms, although the suggestion is generally constructive, changing the gap scenarios does not seem to add that much of a value to this paper. Nonetheless, I am happy to redo the study with the suggested approach if the reviewer or the editor insists on that.

RC01: Moffat et al. (2007) concluded that the quite simple and widely applied MDS algorithm for filling Fc, i.e. NEE time series, which is using only the common variables NEE, Rg, Tair, and VPD as predictors. What are the reasons to omit this for many sites "business as usual"-algorithm? The computation can even be outsourced to the online tool provided by the MPI-BGC Jena.

AC: We accept the suggestion, and we would include the MDS.

RC01: P7L226: Were all the eight drivers used or a subset of them, maybe different by method? What is q? The formulation "by trial and error" needs more explanation.

AC: All the eight drivers were used for all methods. Symbol q is the specific humidity, which should have mentioned on table 2. Here "trial and error" was made based on applying feature importance analysis using random forest, and then feeding the algorithms with the different combinations of the suggested features to find out which combination provide the best performance metrics. We will explain that in the revised version.

RC01: P10L308: Here it does not become clear what cross-sections have been used. I imaged some categories based on similar environmental conditions or day/night time. This only becomes clear in the discussion, in that data from other sites have been used with site as cross-section. This cross-site gap-filling is hard to transfer to other studies. In what respect does the PD model differ from a classical mixed effects models?

AC: For each tower, we used the four rest towers as its cross-sections. Now that we know how much important the similarity of the cross-sections are, it is obvious that the method can be used for the regions where the density of towers are high enough, e.g. central Europe. Nonetheless, the computational problem is also a big concern, making the method not to be feasible, at least as long as our computational power has not been dramatically changed. Regarding the difference of PD from classical mixed effect models, it should be noticed that PD can be considered as a combination of a classical mixed effect model with a time series model ,e.g. ARIMA models. More information will be provided in the methods, accordingly.

RC01: P27L720 Conclusions 4 and 5 are mere speculations given the results presented in the paper. They should be moved to the discussion. Contrary to the suggestion 4, I hypothesize that using net radiation as a predictor should handle this case already well (at least with RF). Otherwise, I suggest first trying to add a nighttime/daytime flag to the set of predictors before splitting the dataset.

AC: We would move conclusions 4 and 5 to the discussion. As for the reviewer's hypothesis, it is a good idea to be tested out. However, as mentioned earlier, this study is the first ongoing series of papers the corresponding author is going to prepare for his thesis. Thus, it is a good idea to include the reviewer's hypothesis in the second paper, since the later should be the logical consequence of what has been found in the

first paper.

RC01: P1L35: Currently, I was confused reading the abstract. It was hard for me to spot the distinction between filling of environmental drivers and filling of fluxes. This can be formulated more clearly.

AC: We do agree with the suggestion. The abstract would be revised thoroughly.

Technical corrections:

RC01: P2L43: This formulation does not become clear to me.

AC: Right point. The sentence is needed to be edited.

RC01: Tab 2: I suggest indicating the commonly used abbreviation for the fluxes in parentheses in addition to the notation of the paper (NEE, LE, H). P11L331: typo: "non-periodic" eq 12: one bar too much.

AC: We do agree.

RC01: P23L583: I suggest to provide another table with method abbreviations or repeat the abbreviations at the beginning of the discussion. By this way you do not force your readers to study the methods section first.

AC: Sounds useful. This will be done.

---

## Author Comment (AC2) · 10 Dec 2020

GENERAL COMMENTS

RC02: The paper presents a detailed evaluation of eight algorithms for gap-filling time series data, using eddy covariance data as a target for the comparisons. The content about the algorithms and the metrics for comparisons are a strong feature of the paper. However, it is more limited in advancing the knowledge of best practices for eddy covariance and micro-meteorological data gap-filling. In other words, the evaluation of the algorithms against each other is of interest, but the chosen test domain is not clearly impacted. It seems that to really benefit the knowledge of methods for gap-filling eddy covariance data, longer time series and more representative gap scenarios would

be necessary, as well as a clear comparison to more established methods. Multi-year datasets are key to properly evaluate these algorithms. Such datasets are now widely available, so it is unclear why only 2013 was used. With this aspect in mind, it seems clear that in the evaluation of the first objective of the paper longer gaps led to disproportional increases in uncertainty. This might not have happened if other years without gaps for the same season were available, for instance. More direct comparisons to "classic" gap-filling algorithms would have helped in this evaluation. Implementations of algorithms such as MDS are now widely available, including as part of OZFlux's own OzFluxQC software package. The comparison of newer methods is informative, but unless compared to currently used solutions, it's hard to assess the improvement. Although the authors are correct, and performance of the MDS algorithm was shown to be comparable to ANNs before, parameterizing MDS is much simpler (no choices in layers, nodes, iterations, or window sizes) and would lead to a more robust and clear comparison.

AC: Please note that as a PhD student whose thesis is based on a series of papers, the current paper is the very first one that has mainly provided as the initial attempt to find out how different algorithms would perform against each other. As such, almost all the points mentioned in the general comment, which are helpful, would be covered in the second paper, e.g. including multiple-year datasets, and applying different random gap scenarios. However, as the second referee has mentioned, we accept the idea of adding the results of the MDS in the current study. Last but not least, the year 2013 was chosen for the fact that the data during the period had less missing data, and that year was a common year of available data amongst all five towers that their data were used. Besides, most of the researches have been done in the field includes just one or two years of data, so the results of this paper can be compared with the majority of similar previous researches. Besides, many researchers still fill the annual gaps by using only the data of that year, thus using a year of data for training the algorithms can still be justified.

RC02: Should the authors choose to really focus on the comparison among the methods presented, I would suggest adding all the comparison metrics RMSE, R2, MBE, etc., for all sites individually and combinations thereof as supplementary materials, making this a valuable and thorough comparison of methods, and reducing the focus from the application to eddy covariance. If the intention really is to show the impact on EC, longer time series and more direct comparisons to current methods would be necessary.

AC: We are happy with adding all the comparison metrics for all sites as supplementary materials. Besides, the intention was to make a comparison between different algorithms, and as such, in case using a year of data is insufficient, it would be equally insufficient for all algorithms.

SPECIFIC COMMENTS

RC02: On the ancillary datasets, it seems they introduce some entanglement to this evaluation. One of the key advantages of purely empirical methods, such as the ones presented in the paper, is that they will not be biased by predefined models (like the reanalysis datasets) or atmospheric interferences (like the MODIS data). After an evaluation without datasets such as these, adding them to improve the methods would be a natural choice. However, without the unbiased evaluation it is hard to qualify the sources of uncertainty in the paper's evaluation.

AC: Even though this is true, the ancillary data used in the current study have been used to gap-fill the drivers' data, and not the fluxes directly. As such, it might not be a concern. Nonetheless, since the corresponding author is going to use the EVI data as one of the gap-filling features for Fc, the corresponding author wonders whether the referee has any suggestion to address the issue?

RC02: Although the performance criteria selected for the paper work well, it is curious to see that the methods all seem to represent high variabilities but fail to capture the extremes, as the authors point out for CO2 and latent heat fluxes – and this doesn't

seem to be the case for sensible heat flux. Could this be an issue of the underlying data requiring further quality control before the gap-filling methods are applied? Or maybe this is an artifact of the period selected in the examples?

AC: This was one of the surprising things raised during the study, and to be honest, we do not have a solid answer to that yet. However, estimating the sensible heat flux is an easier task as against the two others. This can justify the exception of sensible heat flux. For Fc, and Fe, our best guess is that the issue happens due to lack of information (hidden features). We will try to figure that out in the second paper of this series.

RC02: The following claim requires either more details or a reference, otherwise it's not possible to know what concerns/challenges the authors are referring to and what aspects of gap-filling the paper is aiming to address: "...there are some serious concerns regarding the challenges associated with the technique, e.g. data gaps and uncertainties."

AC: That is right. The relevant references, as well as more explanation, will be added to clarify the sentence.

RC02: The +/-25gCm-2y-2 (Moffat et al. 2007) and +/-30gCm-2y-2 (Richardson & Hollinger 2007) are dependent on the underlying datasets used for the evaluation. These numbers should not be taken as general benchmarks.

AC: That is the right point. This part will be modified, and an appropriate reference will be added.

RC02: In the sentence "Nevertheless, one of the concerns regarding this algorithm is that the independent variables, here meteorological drivers, might be auto-correlated." it is unclear why this would be a concern, since the meteorological drivers being auto-correlated is one of the assumptions that allow the MDS method to work.

AC: The comment is true. We will delete the sentence.

RC02: The sentence "This challenge becomes acute when the gaps happen within

a period when the ecosystem behaviour is changing and thereby showing different response under similar meteorological conditions." is another reason why multi-year datasets should be used to compare these algorithms.

AC: As mentioned earlier, we used one year of data because: (a) the focus had been on algorithm comparison, (b) most previous researches used a year or two, so our results could be more comparable with them, and (c) 2013 was the year during which the datasets for all five towers were smaller proportions of gaps. Finally, in the second paper of the series, we are using up to five years of data for training and testing. Hence, the concern would be considered in the bigger picture.

RC02: The gap scenarios and training windows selected are highly structured and rigid. It's unclear how the evaluation over these scenarios would translate into real-world examples, which have both structured gaps (e.g., from sensor failures) and arbitrary gaps (e.g., from data filtering). It seems it would be important to use are least one scenario with gaps and training data both randomized, and also combinations of lengths for gap windows and training windows.

AC: This is a good and constructive suggestion. However, please note that this paper is the first one of a series of papers the corresponding author has to write down as his PhD thesis. For the second one, which the corresponding author is working on right now, randomly selected gaps have been superimposed.

TECHNICAL CORRECTIONS

— Abstract -

RC02: The acronyms RF and CLR were referenced before being defined

AC: Thank you for letting us know that. Those acronyms would be predefined in the revised version.

RC02: - "...RF provided more consistent results with less bias, relatively." It would be clearer to describe "relatively" to what in this sentence.

AC: That is a helpful suggestion. The authors mean related to the rest of the algorithms used in the study. The sentence would become modified accordingly.

RC02: - This sentence is a bit unclear "In each scenario, the gaps covered the data for the entirety of 2013 by consecutively repeating them, where, in each step, values were modelled by using earlier window data." Were measured and modelled data used simultaneously in training? — Introduction

AC: The reviewer has pointed out an important issue that is the explanation of the scenarios is not clear enough, particularly because this was the issue for the first reviewer as well. As such, we need to rewrite the sentence to make sure that the method is clear enough and easy to understand.

RC02: - "...and not measured at the point." Maybe could be "not measured at a point scale"?

AC: That is right. We will correct it.

RC02: - A more classic reference for FLUXNET is: Baldocchi et al. 2001. FLUXNET: A New Tool to Study the Temporal and Spatial Variability of Ecosystem-Scale Carbon Dioxide, Water Vapor, and Energy Flux Densities. BAMS, 11: 2415-2434.

AC: Thank you for reminding that reference. We would use include it in the introduction.

RC02: - And more appropriate references for EUROFLUX and AmeriFlux are: Aubinet, M. et al. 1999. Estimates of the Annual Net Carbon and Water Exchange of Forests: The EUROFLUX Methodology. Advances in Ecological Research, pp. 113–175. Law, B. 2007. AmeriFlux Network aids global synthesis. Eos, 88, 286–286. Novick, K. A. et al 2018. The AmeriFlux network: A coalition of the willing. AFM, 249:444-456.

AC: We would consider the references.

RC02: - "Despite the capability of EC to frequently validate process modelling analyses..." might be more precisely phrased as something like "Despite EC data being

frequently used to validate process modelling analyses..."

AC: Yes. That would make the sentence more natural and smooth.

RC02: - "[...] Moffat et al. (2007) compared a couple of different commonly-used gap-filling algorithms"; in fact, Moffat et al. 2007 compared 15 gap-filling techniques.

AC: Yes, we would mention the exact number, instead of "a couple".

- Materials and Methods

RC02: - "and Tumbarumba form 2011 to 2013..." form -> from

AC: Thank you for mentioning the mistake.

RC02: - "Each algorithm was tuned up individually using gird search,..." gird -> grid

AC: Thank you for mentioning the mistake.

— Results

RC02: - Even with a maximum zoom in the PDF file, it is rather hard to read the axis for Figures 3 and 4

AC: That happens because the size/resolution of the figures is not big/high enough. The figures would be replaced with appropriate ones.

— Discussion

RC02: - This sentence is unclear: "That is because ANNs have been checking out for a long time in different locations and considered as one of the most reliable algorithms in the field for more than a decade"

AC: The authors mean occasional superiority of random forest algorithm, needs to be happen in several future researches to convince us to suggest RF instead of ANNs, or identify the algorithm as another standard method. We will add a sentence to clarify the point.

---

## Author Response (AR1)

**Thomas Wutzler (Referee)** twutz@bgcjena.mpg.de

**General comments**

The paper by Mahabbati et al. presents an updated comparison of gap-filling algorithm, which are an important tool in the analysis of data from eddy-covariance sensors and understanding the ecosystem functioning. Their methodology is oriented at the Australian version of the data processing chain taking into account information in addition to the eddy-stations from weather forecasting models and from BIOS2 model data integration environment. For gap-filling of meteorological drivers, they corroborate previous findings of complex methods being not much better than simple methods. Contrary, for the carbon fluxes itself they find a better performance of the machine learning (ML) based approaches. This study is a valuable contribution to the Australian setup. However, their findings are difficult to transfer to other processing setups and other sites. Hence, the paper is a quite special application and in the current form better suited for an Australian journal.

Even though the data used in this paper came from Australia, the focus was to find out whether ML algorithms other than ANNs can provide more robust results regarding gap-filling of drivers and fluxes. That being said, the towers are selected just as samples to compare the performance of different algorithms. In that sense, the paper is algorithm-oriented rather than Australian-style oriented, and the output is suitable for all members of the FLUXNET. Please note that the diversity amongst the towers has been wide, and there is less likely that an algorithm like RF, which has consistently provided a robust performance in all the five different climates and sites, perform poorly in other parts of the world, or with different input features. The initiation of this study is to compare different algorithms.

I encourage the authors for a major revisions to extent their study to setups that are comment also applicable at other sites for submission to GI.

I have several major concerns, which I state here and explain below. First, I propose to add a comparison with fitting the models to only data that are commonly available at other sites. Second, the methodology needs to be updated to introduce gaps at random positions in time instead of all starting at 1st of January to avoid confounding of gap-length with seasonality. Third, I propose to include the MDS algorithm that was simple but well performing at previous gap-filling comparisons and a "business as usual" for gap-filling NEE at many sites.

For the first suggestion, since the main goal of the study was to compare different gap-filling algorithms, we do not believe changing the input data leads to a difference in the relative performance

of the algorithms. Moreover, as mentioned in the materials and methods, a variety of climates is involved in this study (Beringer et al. 2016), which makes the results useful for different types of audiences. Australia's area is almost as twice as big as Western Europe, and it has a large variety of climates.

For the second suggestion, we accepted and changed the gap-filling scenario. The gaps have now been selected randomly during 2013. The gap scenario part has been changed to: *"In order to find out the effect of gap size on the performance of our gap-filling algorithms, the data of nine different gap windows (i.e. 1, 5, 10, 20, 30, 60, 90, 180 and 365 consecutive days) were removed randomly from the datasets during 2013. Afterwards, the data from 2011 to 2013 were used to train the algorithms. Finally, the trained algorithms were used to fill the artificial gaps superimposed to the datasets. The entire process permutated five times in each scenario to ensure the performance was not sensitive to the gap period. As such, 15 variables, 9 window lengths, 8 gap-filling methods (MDS excluded), and 5 permutations across 5 towers resulted in 27000 computations for the meteorological features. Similarly, 3 fluxes, 9 window lengths, 9 gap-filling methods, and 5 permutations across 5 towers resulted in 6075 computations for the major fluxes, overall."*

For the third suggestion, we have included the MDS method for the major fluxes.

**Specific comments**

In order to be usable at other sites, the methods should be compared in addition to the presented setup by using only data commonly available at eddy-covariance sites, which are the measurements themselves (Fc, Fh, Fe) together with ancillary measurements (Rg, VPD, rH, Tair, Tsoil, Ustar, precip, wind speed, and wind direction), and maybe another comparison using in addition more detailed radiation measurements and ground heat flux and soil water storage (Table 2).

We believe that it is a useful suggestion. However, as mentioned earlier, the main goal was to compare different gap-filling algorithms and it is less likely that changing the input features makes any change in the comparative performance of the models. For instance, Kim (Kim et al 2019) compared ANNs, RF, SVR and MDS to fill the gaps of Methane flux with different input features than this study, and the performance ranking amongst the ML methods was quite similar to this paper: RF outperformed the rest, and ANNs outperformed SVR. Besides, the data used in that research came from North America. Nonetheless, we included the MDS method using the commonly used input features: Fsd, Ta, and VPD.

In the current comparison setup, the larger gap-lengths comprise a larger proportion of other seasons, while the short gap-lengths only comprise summer records. Hence, the conclusions on gap-lengths are confounded with seasonality. I suggest to randomly distribute gaps in the portion of the entire data series with sufficiently high proportion of non-missing original data. Moreover, most data-processing setups will not fit a model for each gap tailored at the gap-length. Hence, I suggest to introduce several gaps (of a given length) across the entire dataset (say of proportions of 40% and 70% of the data according to p6L215) and let each methods fill all these gaps and compute the statistics across all the gaps but also of the aggregated annual value. In this way a recommendation can be presented that is closer to the gap-filling as applied at many sites. The decision to adjust the training window to the gap-length is very difficult to compare to other gap-filling of real time series where gap-lengths vary. Most investigators will not effort to fit a model around each gap. I suggest

training the methods on a shifting window and filling all gaps inside this window, and for efficiency use only few increasing window lengths of the training.

This is a good suggestion, and this is a better approach in general for a realistic gap-filling process. As such, we changed the gap-filling scenario to the following: for each gap length, we randomly picked out a period and removed the data. Then we trained the algorithms with the rest of data, and filled the gaps. The entire process permutated five times in each scenario to ensure the performance was not sensitive to the gap period. However, the gaps were chosen consecutively to be more challenging for the algorithms. Short gaps have not been considered a concern, overall.

Moffat et al. (2007) concluded that the quite simple and widely applied MDS algorithm for filling Fc, i.e. NEE time series, which is using only the common variables NEE, Rg, Tair, and VPD as predictors. What are the reasons to omit this for many sites "business as usual"-algorithm? The computation can even be outsourced to the online tool provided by the MPI-BGC Jena.

We accepted the suggestion and included the MDS.

P7L226: Were all the eight drivers used or a subset of them, maybe different by method? What is q? The formulation "by trial and error" needs more explanation.

All the eight drivers were used for all methods, except for the FBP and MDS. Symbol q is the specific humidity, which has now been mentioned on table 2. Here "trial and error" was made based on applying feature importance analysis using random forest, and then feeding the algorithms with the different combinations of the suggested features to find out which combination provide the best performance metrics. The sentence has been edited like this: *"... based on a combination of RF feature selection and testing out a series of feature combinations."*

P10L308: Here it does not become clear what cross-sections have been used. I imaged some categories based on similar environmental conditions or day/night time. This only becomes clear in the discussion, in that data from other sites have been used with site as cross-section. This cross-site gap-filling is hard to transfer to other studies. In what respect does the PD model differ from a classical mixed effects models?

For each tower, we used the four rest towers as its cross-sections. Now that we know how much important the similarity of the cross-sections are, it is obvious that the method can be used for the regions where the density of towers are high enough, e.g. central Europe. Nonetheless, the computational problem is also a big concern, making the method not feasible, at least as long as our computational power has not been dramatically changed. Regarding the difference of PD from classical mixed effect models, it should be noticed that PD can be considered as a combination of a classical mixed effect model with a time series model ,e.g. ARIMA models. The additional cross-sections information has been provided in the methods, accordingly.

P27L720 Conclusions 4 and 5 are mere speculations given the results presented in the paper. They should be moved to the discussion. Contrary to the suggestion 4, I hypothesize that using net radiation as a predictor should handle this case already well (at least with RF). Otherwise, I suggest first trying to add a nighttime/daytime flag to the set of predictors before splitting the dataset.

We merged the conclusions 4 and 5, and moved them to the discussion. As for the reviewer's hypothesis, it is a good idea to be tested out. However, as mentioned earlier, this study is the first ongoing series of papers the corresponding author is going to prepare for his thesis. Thus, it is a good

idea to include the reviewer's hypothesis in the second paper, since the later should be the logical consequence of what has been found in the first paper.

P1L35: Currently, I was confused reading the abstract. It was hard for me to spot the distinction between filling of environmental drivers and filling of fluxes. This can be formulated more clearly.

The abstract has been revised thoroughly to address the issue.

**Technical corrections**

P2L43: This formulation does not become clear to me.

Right point. The sentence has been edited.

Tab 2: I suggest indicating the commonly used abbreviation for the fluxes in parentheses in addition to the notation of the paper (NEE, LE, H). P11L331: typo: "non-periodic" eq 12: one bar too much.

All suggestions have been done.

P23L583: I suggest to provide another table with method abbreviations or repeat the abbreviations at the beginning of the discussion. By this way you do not force your readers to study the methods section first.

Sounds useful. This has been done.

**The Second Reviewer:**

**Anonymous Referee #2**

**GENERAL COMMENTS**

The paper presents a detailed evaluation of eight algorithms for gap-filling time series data, using eddy covariance data as a target for the comparisons. The content about the algorithms and the metrics for comparisons are a strong feature of the paper. However, it is more limited in advancing the knowledge of best practices for eddy covariance and micro-meteorological data gap-filling. In other words, the evaluation of the algorithms against each other is of interest, but the chosen test domain is not clearly impacted. It seems that to really benefit the knowledge of methods for gap-filling eddy covariance data, longer time series and more representative gap scenarios would be necessary, as well as a clear comparison to more established methods. Multi-year datasets are key to properly evaluate these algorithms. Such datasets are now widely available, so it is unclear why only 2013 was used. With this aspect in mind, it seems clear that in the evaluation of the first objective of the paper longer gaps led to disproportional increases in uncertainty. This might not have happened if other years without gaps for the same season were available, for instance. More direct comparisons to "classic" gap-filling algorithms would have helped in this evaluation. Implementations of algorithms such as MDS are now widely available, including as part of OZFlux's own OzFluxQC software package. The comparison of newer methods is informative, but unless compared to currently used solutions, it's hard to assess the improvement. Although the authors are correct, and performance of the MDS algorithm was shown to be comparable to ANNs before, parameterizing MDS is much simpler (no choices in layers, nodes, iterations, or window sizes) and would lead to a more robust and clear comparison.

Please note that as a PhD student whose thesis is based on a series of papers, the current paper is the very first one that has mainly provided as the initial attempt to find out how different algorithms would perform against each other. As such, almost all the points mentioned in the general comment, which are helpful, would be covered in the second paper, e.g. including multiple-year datasets, and applying different random gap scenarios. However, as the second referee has mentioned, we accept the idea of adding the results of the MDS in the current study. Last but not least, the year 2013 was chosen for the fact that the data during the period had less missing data, and that year was a common year of available data amongst all five towers that their data were used. Besides, most of the researches have been done in the field includes just one or two years of data, so the results of this paper can be compared with the majority of similar previous researches. Besides, some researchers still fill the annual gaps by using only the data of that year, thus using a year of data for training the algorithms can still be justified.

Should the authors choose to really focus on the comparison among the methods presented, I would suggest adding all the comparison metrics RMSE, R2, MBE, etc., for all sites individually and combinations thereof as supplementary materials, making this a valuable and thorough comparison of methods, and reducing the focus from the application to eddy covariance. If the intention really is to show the impact on EC, longer time series and more direct comparisons to current methods would be necessary.

We are happy with adding all the comparison metrics for all sites as supplementary materials. Besides, the intention was to make a comparison between different algorithms, and as such, in case using a year of data is insufficient, it would be equally insufficient for all algorithms.

**SPECIFIC COMMENTS**

On the ancillary datasets, it seems they introduce some entanglement to this evaluation. One of the key advantages of purely empirical methods, such as the ones presented in the paper, is that they will not be biased by predefined models (like the reanalysis datasets) or atmospheric interferences (like the MODIS data). After an evaluation without datasets such as these, adding them to improve the methods would be a natural choice. However, without the unbiased evaluation it is hard to qualify the sources of uncertainty in the paper's evaluation.

Even though this is true, the ancillary data used in the current study have been used to gap-fill the drivers' data, and not the fluxes directly. As such, it might not be a concern.

Although the performance criteria selected for the paper work well, it is curious to see that the methods all seem to represent high variabilities but fail to capture the extremes, as the authors point out for CO2 and latent heat fluxes – and this doesn't seem to be the case for sensible heat flux. Could this be an issue of the underlying data requiring further quality control before the gap-filling methods are applied? Or maybe this is an artifact of the period selected in the examples?

This was one of the surprising things raised during the study, and to be honest, we do not have a solid answer to that yet. However, estimating the sensible heat flux is an easier task as against the two others. This can justify the exception of sensible heat flux. For Fc, and Fe, our best guess is that the issue happens due to lack of information (hidden features). We will try to figure that out in the second paper of this series.

The following claim requires either more details or a reference, otherwise it's not possible to know what concerns/challenges the authors are referring to and what aspects of gap-filling the paper is aiming to address: "...there are some serious concerns regarding the challenges associated with the technique, e.g. data gaps and uncertainties."

Those concerns have been explained in the following paragraphs.

The +/-25gCm-2y-2 (Moffat et al. 2007) and +/-30gCm-2y-2 (Richardson & Hollinger 2007) are dependent on the underlying datasets used for the evaluation. These numbers should not be taken as general benchmarks.

That is the right point. The point has been emphesised in order to not mislead the reader.

In the sentence "Nevertheless, one of the concerns regarding this algorithm is that the independent variables, here meteorological drivers, might be auto-correlated." it is unclear why this would be a concern, since the meteorological drivers being autocorrelated is one of the assumptions that allow the MDS method to work.

The comment is true. We have deleted the sentence.

The sentence "This challenge becomes acute when the gaps happen within a period when the ecosystem behaviour is changing and thereby showing different response under similar meteorological conditions." is another reason why multi-year datasets should be used to compare these algorithms.

Firstly, the gap-filling scenario has been changed in such a way that the data of up to two years (2012 and 2013) have been used now. Secondly, as mentioned earlier, we have not used multiple years of data because: (a) the focus had been on algorithm comparison, (b) most previous researches used a year or two, so our results could be more comparable with them, and (c) 2013 was the year during which the datasets for all five towers were smaller proportions of gaps. Finally, in the second paper of the series, we are using up to five years of data for training and testing. Hence, the concern would be considered in the bigger picture.

The gap scenarios and training windows selected are highly structured and rigid. It's unclear how the evaluation over these scenarios would translate into real-world examples, which have both structured gaps (e.g., from sensor failures) and arbitrary gaps (e.g., from data filtering). It seems it would be important to use are least one scenario with gaps and training data both randomized, and also combinations of lengths for gap windows and training windows.

This is a good and constructive suggestion. we accepted and changed the gap-filling scenario. The gaps have now been selected randomly during 2013. The gap scenario part has been changed to: *"In order to find out the effect of gap size on the performance of our gap-filling algorithms, the data of nine different gap windows (i.e. 1, 5, 10, 20, 30, 60, 90, 180 and 365 consecutive days) were removed randomly from the datasets during 2013. Afterwards, the data from 2011 to 2013 were used to train the algorithms. Finally, the trained algorithms were used to fill the artificial gaps superimposed to the datasets. The entire process permutated five times in each scenario to ensure the performance was not sensitive to the gap period. As such, 15 variables, 9 window lengths, 8 gap-filling methods (MDS excluded), and 5 permutations across 5 towers resulted in 27000 computations for the meteorological features. Similarly, 3 fluxes, 9 window lengths, 9 gap-filling methods, and 5 permutations across 5 towers resulted in 6075 computations for the major fluxes, overall."*

**TECHNICAL CORRECTIONS**

**— Abstract -**

The acronyms RF and CLR were referenced before being defined

Thank you for letting us know that. Those acronyms have been predefined in the revised version.

- "...RF provided more consistent results with less bias, relatively." It would be clearer to describe "relatively" to what in this sentence.

That is a helpful suggestion. The authors mean related to the other ML algorithms used in the study. The sentence has been edited.

- This sentence is a bit unclear "In each scenario, the gaps covered the data for the entirety of 2013 by consecutively repeating them, where, in each step, values were modelled by using earlier window data." Were measured and modelled data used simultaneously in training? — Introduction

The scenario has changed, and so as the mentioned quotation.

- "...and not measured at the point." Maybe could be "not measured at a point scale"?

That is right. We edited the sentence.

- A more classic reference for FLUXNET is: Baldocchi et al. 2001. FLUXNET: A New Tool to Study the Temporal and Spatial Variability of Ecosystem-Scale Carbon Dioxide, Water Vapor, and Energy Flux Densities. BAMS, 11: 2415-2434.

Thank you for reminding that reference. We would use include it in the introduction.

- And more appropriate references for EUROFLUX and AmeriFlux are: Aubinet, M. et al. 1999. Estimates of the Annual Net Carbon and Water Exchange of Forests: The EUROFLUX Methodology. Advances in Ecological Research, pp. 113–175. Law, B. 2007. AmeriFlux Network aids global synthesis. Eos, 88, 286–286. Novick, K. A. et al 2018. The AmeriFlux network: A coalition of the willing. AFM, 249:444-456.

We used one of the references in the revised version.

- "Despite the capability of EC to frequently validate process modelling analyses..." might be more precisely phrased as something like "Despite EC data being frequently used to validate process modelling analyses..."

The suggestion has been considered.

- "[...] Moffat et al. (2007) compared a couple of different commonly-used gap-filling algorithms"; in fact, Moffat et al. 2007 compared 15 gap-filling techniques.

Right. We have replaced "15" instead of "a couple".

- **Materials and Methods**

- "and Tumbarumba form 2011 to 2013..." form -> from

Thank you for mentioning the mistake.

- "Each algorithm was tuned up individually using gird search,..." gird ->
grid
Thank you for mentioning the mistake.

— **Results**

-	Even with a maximum zoom in the PDF file, it is rather hard to read the axis for Figures 3 and 4

Since the scenario has changed in the revised version, the mentioned figures could not be plotted anymore. They have been removed.

— **Discussion**

-	This sentence is unclear: "That is because ANNs have been checking out for a long time in different locations and considered as one of the most reliable algorithms in the field for more than a decade"
The authors mean occasional superiority of random forest algorithm, needs to happen in several future studies to convince us to suggest RF instead of ANNs, or identify the algorithm as another standard method. We will add a sentence to clarify the point.

---

## Author Response (AR2)

**Author's response to the reviews**

**Suggestions for revision or reasons for rejection (will be published if the paper is accepted for final publication)**

Submitted on 16 Feb 2021
Reviewer #1: Thomas Wutzler, twutz@bgc-jena.mpg.de

General comments

The paper by Mahabbati et al. presents an updated comparison of gap-filling algorithm, which are an important tool in the analysis of data from eddy-covariance sensors and understanding the ecosystem functioning. Their methodology is oriented at the Australian version of the data processing chain taking into account information in addition to the eddy-stations from weather forecasting models and from BIOS2 model data integration environment. They corroborate previous findings of complex methods being not much better than simple methods for gap-filling of meteorological drivers. Contrary, for the carbon fluxes itself they find a better performance of the machine learning (ML) based approaches.

The current revision took into account two of my three major concerns (gaps vs. seasonality and adding MDS). However, their comparison of gap-filling still only uses a combination of drivers that is quite specific to the Australian setup (weather-forecasting model, and the Australian BIOS2 model-data-integration environment) and hard to transfer to other sites and setups. The inclusion of the ancillary datasets is (no doubt) valueable for the performance of the gapfilling. But including an additional comparison-scenario with a constraint set of drivers to my opinion would greatly help the transferability of conclusions on the choice of methods to other readers. Hence, I still encourage the authors to include such a scenario. Nevertheless, given that the usage of quite the specific set of drivers is made clear enough, the study is worth publishing without this additional scenario.

*We thank the first reviewer for his constructive comments. Using the common drivers for gap-filling the fluxes would definitely increase the transferability of the conclusions. Since the corresponding author needs to write two other papers for his PhD completion, we would consider the third concern in the upcoming papers.*

I congratulate the authors to achieve running all these various types of approaches in the same setup on the same dataset.
*We appreciate the time and effort the first reviewer has put in reviewing this paper. Moreover, the corresponding author wants to thank the first reviewer for introducing REddyProc online tool, which ha has found easy-to-use and convenient.*

Specific comments

My concerns about "Australian setup" did not concern the selection of sites, but rather the selection of the set of inputs to the gap-filling which maybe not available at other sites.
*Now that we understand the concern we would consider it for the next paper of the corresponding author's series of papers for his PhD. That is a key point for generalising the conclusions to the global community. Besides, it is worth mentioning the study undertaken by*

*Moore et al. (2020) where the authors used the "Australian methodology" to gap-fill the flux data (e.g. using ERA-Interim and local weather station data) in the Midwest region of the United States that provided satisfying results.*

To me, the current setup of "gap-filling" of environmental drivers reads more like a downscaling or interpolation/integration of various sources. The same variable from various sources is used as a driver for the prediction this variable.

The authors claim in their reply to my specific comments: "it is less likely that changing the input features makes any change in the comparative performance of the models." I am not in a position to asses this claim.

However, this claim together with summarizing the specific drivers should be placed prominently in the discussion together with the citations given in their reply to my concern. From my previous report I repeat my suggestion of an additional scenario "using only data commonly available at eddy-covariance sites, which are the measurements themselves (Fc, Fh, Fe) together with ancillary measurements (Rg, VPD, rH, Tair, Tsoil, Ustar, precip, wind speed, and wind direction)". Then you can also compare the very common case of using MDS of filling Tair.

*First, we would like to clarify that in the PyFluxPro, the suite of scripts whereby the EC data are processed in the OzFlux Network, just one of the ancillary sources is used to gap-fill each meteorological driver depending on availability of the data based on a priority. In this study, however, more than one source of data are used together to fill the drivers' gaps since the outcome provided lower values for the RMSE. The sentence "it is less likely that changing the input features makes any change in the comparative performance of the models." refers to gap-filling of fluxes based on the drivers some of which might commonly been used as input data for the gap-filling process.*
*We have included the suggestion in the discussion part, and the corresponding author would consider using the commonly available drivers in his following paper of the series.*

Thanks for adopting the suggestion of the distribution of the consecutive gaps and the fitting to the entire data. Please, also state this also in the manuscript (section 2.3?). Currently, that way of training the model (with the data from 2012 and the data from 2013 excluding the artificial gaps, correct?), does not become clear in the current version of the manuscript.
*That has been addressed in section 2.3.*

Minor comments / Technical corrections

Tables: I found it hard to keep associating values within the same row. Please, consider adding some horizontal guiding lines.
*Horizontal guiding lines have been added.*

Table 3: Please, link to the text where the data-sources are described and maybe provide a summary in the table caption.
*The table has been linked to the text.*

I found it hard to always switch back to Table 2. Please, consider repeating the meaning of some acronyms at the relevant paragraphs, e.g. heading 3.1.1 "CO2 flux (FC)". I would prefer some slightly longer acronyms, e.g. Tair, Tsoil compared to Ta and Ts.

*The meaning of the ancronyms are added in part 3.*

In the version I got, some reference in parenthesis are missing, e.g. P15L339, or P16L458. *The references have been added accordingly.*

*The references have been added.*

*Moore, C. E., Berardi, D. M., Blanc-Betes, E., Dracup, E. C., Egenriether, S., Gomez-Casanovas, N., Hartman, M. D., Hudiburg, T., Kantola, I., Masters, M. D., Parton, W. J., Van Allen, R., von Haden, A. C., Yang, W. H., DeLucia, E. H. and Bernacchi, C. J.: The carbon and nitrogen cycle impacts of reverting perennial bioenergy switchgrass to an annual maize crop rotation, GCB Bioenergy, 12(11), 941–954, doi:10.1111/gcbb.12743, 2020.*

**Suggestions for revision or reasons for rejection (will be published if the paper is accepted for final publication)**

Submitted on 25 Feb 2021
Anonymous Reviewer #2

GENERAL COMMENTS

The two main changes in this version of the paper are the addition of the MDS method for gap-filling fluxes and a randomized selection of introduced gaps. These are both considerable improvements and make the results from the methods comparison more robust. However, the short records of data (only 1-2 years) was also a key concern from both reviewers and was not addressed in this revision. The paper, as it is, represents an interesting contribution showing methods were mostly comparable with the short records and the traditional MDS algorithm still performs reasonably well, and potentially the more complex methods might not lead to improvements that are worth the extra costs. However, these are not conclusive results from the paper. The paper could have been a key contribution to the literature, and although the contributions seem to be technically sound, they do not advance the state of the art in gap-filling of eddy covariance data.

SPECIFIC COMMENTS

In an answer to a reviewer comment, the authors state: "...since the main goal of the study was to compare different gap-filling algorithms, we do not believe changing the input data leads to a difference in the relative performance of the algorithm". For the comparison of these algorithms, the only factor changing their performance will be the input data. The input data is even more important for methods such as ANNs and RF, which are entirely dependent on relationships between the data variables.

*As for the relative performance of the models, it is more likely that as long as all models using the same input features, the relative performance does not change considerably. For instance, Kim et al. (2020) used ANNs, RF, SVR, and MDS to fill the data gaps of methane, and the relative performance of the models almost matches the current study. Additionally, the relative performance of the methods in this study remains similar amongst the five towers, which reinforces the assumption. Finally, for the next paper of this series, we would train the developed gap-filling models using the commonly used drivers, which would address your concern.*

Still in the answers, about ancillary datasets: "Even though this is true, the ancillary data used in the current study have been used to gap-fill the drivers' data, and not the fluxes directly. As such, it might not be a concern." It might be good to clarify in the methods that only the measured values for the drivers were used to gap-fill fluxes. Although it is fair to assume no gap-filled driver data was used to fill the fluxes, I couldn't find this statement in the paper.

*Just to clarify: the ancillary datasets are used to gap-fill the drivers. For gap-filling the fluxes, those gap-filled drivers are used, otherwise the regression methods could not fill the gaps for which they do not have the corresponding drivers. In other words, the drivers used to fill the gaps of fluxes had included both the measured values and the gap-filled values. I wonder how it would be possible for an ANNs model to fill the gaps of a flux while it does not have access to the corresponding input data.*

The argument that many previous research results use only single years for evaluation omits that most of these had limited access to long and uniform records. With record spanning over 20 years of data available from most regional flux networks, this is not a limitation any longer and should have been integral to the paper. Seasonal patterns can be correctly identified by many of the methods used, but only if using multi-year data. Using single year limited the results of the paper, which could have been a considerable contribution to both the eddy covariance and machine learning scientific communities.

*The point is generally true and quite useful. But, we have had some considerations to use a limited years of data as follows:*

*a) The results of this study would still be useful for the newly established EC sites and for the sites which had been active for a short period of time. It would be a valuable knowledge to know how different gap-filling algorithms perform when the training data is timely limited.*

*b) Amongst the algorithms, panel data turned out to be memory-hungry. That being said increasing the training period would have increased the memory demand at a level which it becomes impossible for us to apply it based on the hardware we had access to.*

*c) As the corresponding author is writing down three papers for his PhD thesis, the idea of using longer training periods would be included in the following papers, which would address the concern.*

In Moffat 2007 the RMSE values for the best performing algorithms (mainly ANN variants but also MDS) were consistently under 3.0 gC m-2 d-1. Since these were consistently higher in this manuscript, this might support the argument that there was too little data to train the runs presented in this paper. Since the year selected to perform the tests was very complete, if the short record is not an limitation, as argued by the authors, one could expect these results to be better.

*The point might be true. Although the RMSE values depend on a variety of factors, including the magnitude of the flux values. For instance, in this study, the RMSE values of Alice Springs Mulga (Tropical and Subtropical Desert Climate) were significantly lower than those of Tumbarumba (Oceanic climate). Moreover, the gap lengths here have been mostly by far longer than those were applied by Moffat 2007 (unlike Moffat 2007 here we had gap lengths of 20 days and longer). The longer gaps have been another factor for larger RMSE values. We will check the effects of longer training data out by applying three different training periods (1, 3, and 5 years) in the next paper of the series.*

The introduction of randomized gaps improves the soundness of the results. However, in the methods, it is a bit unclear how all the many realizations of the random gaps were aggregated for the final results. This could be explained in more detail. As an example, it is curious that the RMSE values for Fc at Alice Springs Mulga are so low, yet the R2 values for the site are also low, while for Tumbarumba, the RMSE values are more within the expected ranges while R2 values are also higher.

*We have edited the final paragraph of 2.4. so that the reader clearly understand the way by which we aggregated the results and reported the performance metrics. As for the later point, since we have not included the site-by-site results in the main manuscript, it might sound a bit irrelevant to mention the point and explain the reason (smaller magnitude of Alice Springs' carbon flux has led smaller RMSE, which does not necessarily mean the performance superiority of the models for the Alice Springs data).*

Finally, I will note that I disagree with the last recommendation in the conclusions.

Ensembles are useful when there isn't a "true" value against which one can compare an estimation value. In gap-filling, artificially introducing gaps (original true values) for comparisons allow precise estimations of uncertainty. Using ensembles for gap-filling would introduce unnecessary uncertainty. However, playing to the strengths of each method one can procedurally combining them (e.g., one method for short and one for long gaps) to improve final results without mixed uncertainties.

*This a reasonable concern. The idea of using an [multistage] ensemble model comes from the idea that in the final stage an averaging would be taken of the outcomes of the chosen methods, which makes the results smoother and prevents large fluctuations* (Yang and Browne. (2004). *As such, we believe that there is a chance of declining the uncertainty of the gap-filling by using an ensemble model. Moreover, there are some researches which claims such advantage for the ensemble models, e.g. (Bormann, H. et al, 2006). However, we are not in a position to reject the second reviewer's opinion. Thus, we believe that it is worth applying the method, calculating the uncertainties, and see whether the pros of using the ensemble models outweigh the cons or not. We are already testing the idea for the second paper of the series.*

TECHNICAL CORRECTIONS

- Net ecosystem exchange (NEE) is usually defined as the sum of $CO_2$ turbulent fluxes (commonly represented as Fc) and $CO_2$ storage fluxes (commonly represented as Sc); so the definition in the paper for Fc as equivalent to NEE can be misinterpreted.
*Since we have mentioned "NEE" as an alternative abbreviation for Fc, and considering the context and other explanations, we believe that it would not be a serious problem, particularly for the readers who are familiar with the EC.*

- It might be good to harmonize formatting for Figures 2, 3, and 4.
*The formats have been harmonised.*

- page 15, L449: missing reference "()"
*The reference has been added.*

- page 24, L703: "3)" -> "4)"
*The number has been corrected.*

- From previous review, in the abstract: The acronyms RF and CLR were referenced before being defined
*Full names have been added.*

*References:*

*Bormann, H., Breuer, L., Croke, B., Gräff, T., Hubrechts, L., Huisman, J. A., ... & Seibert11, J.: Reduction of predictive uncertainty by ensemble hydrological modelling of discharge and land use change effects., Uncertainties 'monitoring-conceptualisation-modelling'sequence catchment Res., 133 [online] Available from:*
*https://www.researchgate.net/publication/234016348_REDUCTION_OF_PREDICTIVE_UN CERTAINTY_BY_ENSEMBLEHYDROLOGICAL_MODELLING_OF_DISCHARGE_AND_L*

ANDUSE_CHANGE_EFFECTS (Accessed 26 March 2021), 2006.

Kim, Y., Johnson, M. S., Knox, S. H., Black, T. A., Dalmagro, H. J., Kang, M., Kim, J. and Baldocchi, D.: Gap-filling approaches for eddy covariance methane fluxes: A comparison of three machine learning algorithms and a traditional method with principal component analysis, Glob. Chang. Biol., 26(3), 1499–1518, doi:10.1111/gcb.14845, 2020.

Yang, S. and Browne, A.: Neural network ensembles: combining multiple models for enhanced performance using a multistage approach, Expert Syst., 21(5), 279–288, doi:10.1111/j.1468-0394.2004.00285.x, 2004.